# Diagnostic utility of DNA methylation analysis in genetically unsolved pediatric epilepsies and CHD2 episignature refinement

Sequence-based genetic testing identifies causative variants in ~ 50% of individuals with developmental and epileptic encephalopathies (DEEs). Aberrant changes in DNA methylation are implicated in various neurodevelopmental disorders but remain unstudied in DEEs. We interrogate the diagnostic utility of genome-wide DNA methylation array analysis on peripheral blood samples from 582 individuals with genetically unsolved DEEs. We identify rare differentially methylated regions (DMRs) and explanatory episignatures to uncover causative and candidate genetic etiologies in 12 individuals. Using long-read sequencing, we identify DNA variants underlying rare DMRs, including one balanced translocation, three CG-rich repeat expansions, and four copy number variants. We also identify pathogenic variants associated with episignatures. Finally, we refine the CHD2 episignature using an 850 K methylation array and bisulfite sequencing to investigate potential insights into CHD2 pathophysiology. Our study demonstrates the diagnostic yield of genome-wide DNA methylation analysis to identify causal and candidate variants as 2% (12/582) for unsolved DEE cases.

The developmental and epileptic encephalopathies (DEEs) are the most severe group of epilepsies, defined by frequent epileptiform activity associated with developmental slowing or regression[1]. While each genetic etiology is rare, with more than 825 genes implicated[2], the cumulative incidence of DEEs overall is 1 in 590 children[3]. Currently, de novo, X-linked, or recessively inherited pathogenic germline variants are found in ~ 50% of individuals with DEEs who undergo genetic testing[4]. These are identified by gene panels, exome sequencing (ES), and now, genome sequencing (GS)[5–7]. A smaller subset is explained by copy number variants (CNVs)[8]. Understanding the etiology guides management, such as clinical trial participation, informs accurate reproductive counseling, enables families to join gene-based support groups, and facilitates the development of targeted therapies[9–12]. This, in turn, improves outcomes but is not possible when the etiology is unknown ("unsolved")[13–15].

Epigenetic modifications, which alter the DNA without inherently changing the DNA nucleotide sequence, determine the etiology of some individuals with neurodevelopmental disorders but have not yet been studied in the DEEs. DNA methylation is an essential epigenetic modification that regulates cellular gene expression by adding a methyl ($CH_3$) group to a DNA strand, typically at CpG sites. This can occur through methylation of promoter CpGs, genomic imprinting, and X-chromosome inactivation[16]. Rare epivariants, defined as rare alterations in DNA methylation with or without identified underlying DNA sequence alterations, contribute to human genetic variation[17], but have also been shown to disrupt normal methylation and transcription to cause disease[18,19]. While DNA methylation does not change the DNA sequence itself, epivariants are often perpetrated by underlying *in-cis* DNA changes, such as rare sequence variants, structural alterations, and CG-rich repeat expansions[17] that are difficult to identify by standard sequencing. One example is the methylation of CGG repeats in the 5' untranslated region (5'UTR) of *FMR1* (MIM:309550) that represses gene expression and causes Fragile X syndrome (MIM:300624)[20]. Similarly, hypermethylation of the 5'UTR of Xylosyltransferase 1 (*XYLT1*, MIM:608124), leading to gene silencing, may identify the "missing" allele in the recessive disease Baratela-Scott

e-mail: Bekim.Sadikovic@lhsc.on.ca; Heather.Mefford@stjude.org

syndrome (BSS [MIM:615777])[21]. In both Fragile X and BSS, the aberrant methylation is due to the expansion of a CG-rich repeat that is difficult to reliably detect using short-read sequencing. Rare epivariants, also called rare differentially (hyper- and hypo-) methylated regions (DMRs), are enriched in individuals with neurodevelopmental disorders and congenital anomalies (ND-CA) compared to controls[22].

In contrast to rare DMRs, which represent discrete genomic regions with outlier methylation changes, genome-wide epigenetic profiles identify a collection of distinct individual CpG site methylation changes across the genome. These epigenetic profiles were first implemented for cancer diagnostics with the introduction of the brain tumor classifier in 2018[23]. A growing number of rare diseases exhibit methylation patterns, or "episignatures," in the blood that are reproducible among individuals with pathogenic variants within the same protein domain, gene, or protein complex, yielding highly sensitive and specific biomarkers[24,25]. Since episignatures in diagnostics of rare neurodevelopmental disorders were first clinically validated and implemented with the EpiSign™ assay in 2019[26], episignatures for nearly 70 rare diseases have been published. Episignatures provide strong evidence for genetic diagnosis, regardless of whether an underlying pathogenic DNA variant is identified, and to resolve variants of uncertain significance (VUS). Episignatures have been found for neurodevelopmental disorders where epilepsy is part of the phenotype[25,27–30], but the diagnostic yield for DEEs has not been determined. Furthermore, how these clinically relevant episignatures might be harnessed to inform underlying disease biology and give insights into potential distinct and overlapping pathogenic mechanisms among disorders is just beginning to be explored[31].

Both rare DMRs and episignatures can be detected in peripheral blood samples. Rare DMRs derived from individuals with ND-CA are recapitulated across multiple tissue types, including blood and fibroblasts[22]. Episignature classifiers for rare diseases are trained on data obtained from blood-derived DNA and are, therefore, blood-specific.

Here, we assessed rare outlier DMRs and DNA methylation signatures in peripheral blood-derived DNA from 582 individuals with genetically unsolved DEEs (uDEEs, Fig. 1). We report our methylation array data processing pipeline, MethylMiner[32], which automates quality control, normalization, and implementation of an algorithm that mines rare DNA methylation events[17] in addition to interactive data visualization. Using a combination of short- and long-read sequencing (LRS), we identify variants underlying rare epivariants and episignatures. Finally, we refine the robust episignature for the DEE gene *CHD2* (MIM:602119)[25] to explore how clinically relevant episignatures may give insights into underlying biology. For individuals with uDEEs, we show that rare epivariants and episignatures uncover molecular causes missed using standard sequence-based approaches.

## Results

### Discovery and validation of DMRs

To determine the ability of our analysis pipeline to robustly detect rare, outlier DMRs, we included DNA from six positive controls with genetic alterations: three individuals with heterozygous or homozygous hypermethylation of *XYLT1*, and three individuals (two males and one female) with hypermethylation of *FMR1*. The outlier DMR analysis detected both rare DMRs (Supplementary Fig. 2, Supplementary Data 3A). Additionally, we identified an *XYLT1* heterozygous hypermethylation carrier in our DEE cohort. Targeted X-chromosome analysis in males identified complete methylation at the *FMR1* locus in both Fragile X males compared to the remaining cohort, all of which were completely unmethylated at *FMR1*. *FMR1* hypermethylation was also higher (~75%) in the Fragile X female sample compared to the other females with 25–50% methylation, likely due to random X-inactivation. Thus, our methylation array analysis approach detects outlier DMRs at known disease loci for the autosomes and sex chromosomes.

Next, we assessed outlier DMRs in our cohort of 1194 individuals (582 uDEEs) across 1226 array samples. We predicted $n = 2184$ total DMRs for the autosomes, $n = 49$ DMRs for males on chromosome X, $n = 27$ DMRs for females on chromosome X, and no DMRs on chromosome Y (Supplementary Data 3B, D, F). After accounting for DMRs overlapping across samples (≥ 50% probe overlap in the same direction of DNA methylation hyper- or hypo-methylation), we derived $n = 1545$ unique DMRs for the autosomes (1009 hyper, 536 hypo), $n = 37$ for males on chrX (26 hyper, 11 hypo), and $n = 22$ for females on chrX (14 hyper, 8 hypo) (Supplementary Data 3C, E, G). Of the samples with one or more outlier DMRs, the majority had only a single outlier DMR (Supplementary Fig. 3).

To determine the robustness of our DMR calling algorithm, we (i) assessed the reproducibility of DMR calls in a subset of samples and (ii) performed validation of DMRs using targeted EM-seq (Supplementary Methods). Using replicate array data for 29 individuals, we found that 80% of DMRs were replicated across different batches for an individual (Supplementary Methods). We then used targeted EM-seq, a non-bisulfite approach, to validate a subset of DMRs. We confirmed that our positive control DMRs (*XYLT1* and *FMR1*) could be detected in the targeted EM-seq data (Supplementary Fig. 4). We then validated 29 outlier DMRs by targeted EM-seq in six individuals with uDEEs and four family members (Supplementary Figs. 5, 6, and 12). In addition to DMR validation, targeted EM-seq provides much higher resolution of the extent of differential methylation than the methylation array (e.g. > 80 methylated CpG sites for the *XYLT1* DMR by targeted EM-seq compared to eight representative probes on array; Supplementary Data 4). Thus, we detected and validated outlier DMRs at higher resolution using an orthogonal approach.

### Rare outlier DMRs in uDEEs

We narrowed down outlier DMR calls for individuals with uDEEs to determine high-priority candidates for further study based on DMR recurrence across multiple individuals, population frequency[17], functional annotations (Methods), and manual inspection of DMR plots for each DMR. We identified 12 individuals with uDEEs with one or more rare, potentially disease-associated DMRs and performed follow-up studies (Table 1, Supplementary Data 2A). One individual had multiple DMRs due to a balanced translocation between chrX and chr13, four individuals each had a DMR due to an expanded CG-rich repeat, and seven individuals had DMRs due to underlying CNVs.

### Rare outlier DMR analysis detects hypermethylation of chr13 due to X;13 translocation

One female with the DEE syndrome, epilepsy of infancy with migrating focal seizures (EIMFS), had 26 rare outlier hypermethylated DMRs across chr13 (Fig. 2A, Supplementary Fig. 7), none of which were present in > 23,000 controls[17]. The DMRs were replicated on a second, independent methylation array from the same individual and validated using targeted EM-seq (Supplementary Fig. 6). Methylation array analysis of both parents revealed that all rare hypermethylated DMRs occurred de novo in the proband (Fig. 2B). Whole-genome Oxford Nanopore Technologies (ONT) long-read sequencing also confirmed the hypermethylated DMRs and identified a balanced translocation between chrX and chr13 (Fig. 2C), annotated as 46,XX,t(X;13) (q28;q14.2). The translocation provides a mechanism whereby random X-inactivation induces hypermethylation on the portion of chr13q attached to the large piece of the X chromosome. The translocation breakpoints were confirmed by PCR and Sanger sequencing of peripheral blood-derived DNA as chrX:152,092,342 to chr13:47,005,269 and chr13:47,005,271 to chrX:152,092,344 (GRCh38/hg38). Parental methylation studies and short-read GS confirmed that the translocation occurred de novo, and SNP analysis revealed that the haplotype containing the translocation was paternally derived. The translocation is likely causative in this individual given the de novo occurrence,

absence of clearly pathogenic sequence variants by trio sequence (Supplementary Data 5), and report of a similar translocation in a female individual with intellectual disability and bilateral retinoblastoma[33].

### Rare outlier DMR analysis detects hypermethylation caused by underlying triplet repeat expansions

We detected two individuals with uDEEs and two control individuals with hypermethylation spanning the 5′UTR and intron 1 of the Casein kinase 1 isoform epsilon (*CSNK1E*, MIM:600863, Fig. 3A) gene. Although present in one control and reported in 6/23,116 controls[17], an individual with DEE and probable haploinsufficiency due to a de novo splicing variant (c.885+1 G > A) in *CSNK1E* has been reported[34], suggesting further study is warranted to determine if variation in this gene causes DEE. Segregation analysis revealed that the hypermethylation in one proband was maternally inherited (Family 1, Supplementary Fig. 8), whereas the other arose de novo (Family 2). After validation of hypermethylation

with targeted EM-seq for both probands (Supplementary Fig. 5), long-read sequencing of the proband (genome, ~1500–3000 bp) and mother (targeted, ~1500 bp) from Family 1 and the proband from Family 2 (genome, ~1100–3200 bp) confirmed the presence of an expanded CGG motif in both (Fig. 3C), as previously reported in individuals with hypermethylation of *CSNK1E* at fragile site FRA22A and reduced expression in lymphoblastoid cells[17]. Through GeneMatcher[35], we identified Family 3 consisting of a proband with the same *CSNK1E* hypermethylated DMR and CGG repeat expansion (genome, ~1300–2100 bp) inherited from his mother (genome, ~270–3500 bp), who is mildly affected by learning, speech, and sleep difficulties (Supplementary Phenotype data). Expression analysis in available fibroblasts from Families 2 and 3 showed that individuals with *CSNK1E* hypermethylation had decreased expression of *CSNK1E* compared to hypermethylation-negative controls (Fig. 3B). Analysis using the OUTRIDER algorithm[36] confirmed "drop-out" of *CSNK1E* (ENSG00000213923) expression compared to publicly available fibroblast controls[37] (Fig. 3B,

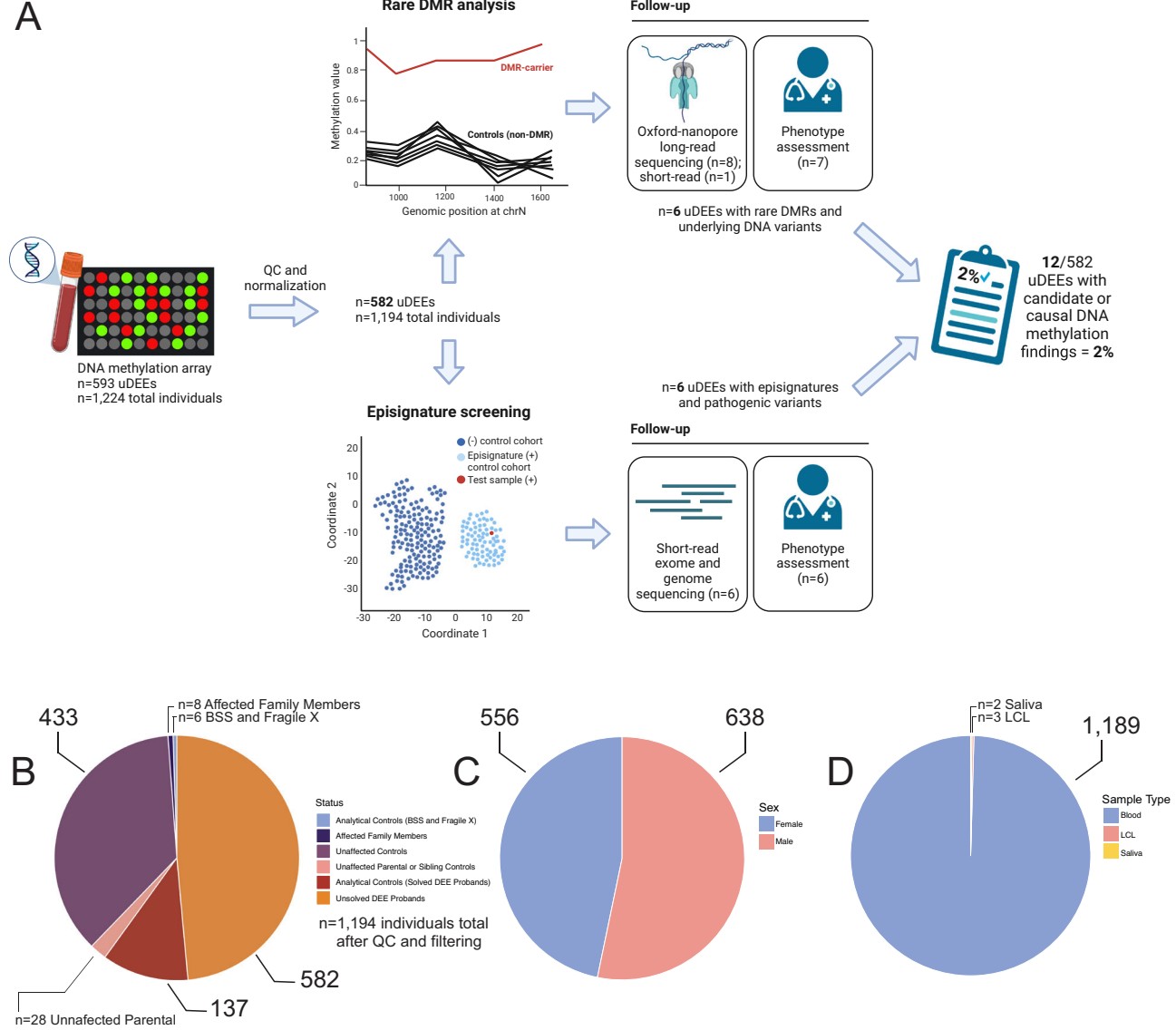

**Fig. 1 | Description of the DNA methylation analysis and features of the study cohort.** **A** Flowchart displaying filtering of samples after quality control (QC) and normalization, analysis pipeline for detecting rare DMRs and episignatures, and narrowing down DNA methylation candidates for this study. **B** Breakdown of the cohort after QC and normalization (*n* = 1194) containing individuals with genetically unsolved DEEs (uDEEs, *n* = 582), unaffected controls (*n* = 461), analytical controls (*n* = 143), and affected family members (*n* = 8). **C** Number of males (*n* = 638) versus females (*n* = 556). **D** Sample type as blood (*n* = 1189) versus other tissue types (*n* = 5). Figure 1A was created with BioRender.com and released under a Creative Commons Attribution-Non-Commercial-Noderivs 4.0 International License (CC-BY-NC-ND).

**Table 1 | Summary of epivariants and underlying DNA sequence alterations identified in this study**

| Location | Gene | Direction | Underlying DNA sequence alteration | Inheritance | Probands (n) |
|---|---|---|---|---|---|
| chr13 | chr13:multiple | Hyper | X;13 translocation (p) | de novo | 1 |
| Xp22 | BCLAF3 | Hyper | CGG repeat (c) | de novo | 1 |
| 22q13 | CSNK1E | Hyper | CGG repeat (c) | Inherited | 2 +1 match |
| 12q13 | DIP2B | Hyper | CGG repeat (c) | Inherited | 1 |
| 16p11.2 | STX1B | Hyper | Deletion (p) | Inherited | 1+3 family |
| 2p16 | CFAP36/CCDC104 | Hyper | Tandem duplication (b) | Inherited | 1 |
| 2q37.3 | chr2:multiple | Hypo | Deletion (b) | Inherited | 1 |
| 15q24 | LINGO1 | Hypo | Deletion (b) | Inherited | 4 |

List of epivariation findings from screening cohort with uDEEs for rare outlier DNA methylation changes. Molecular findings are considered p=pathogenic, c=candidate, and b=benign.

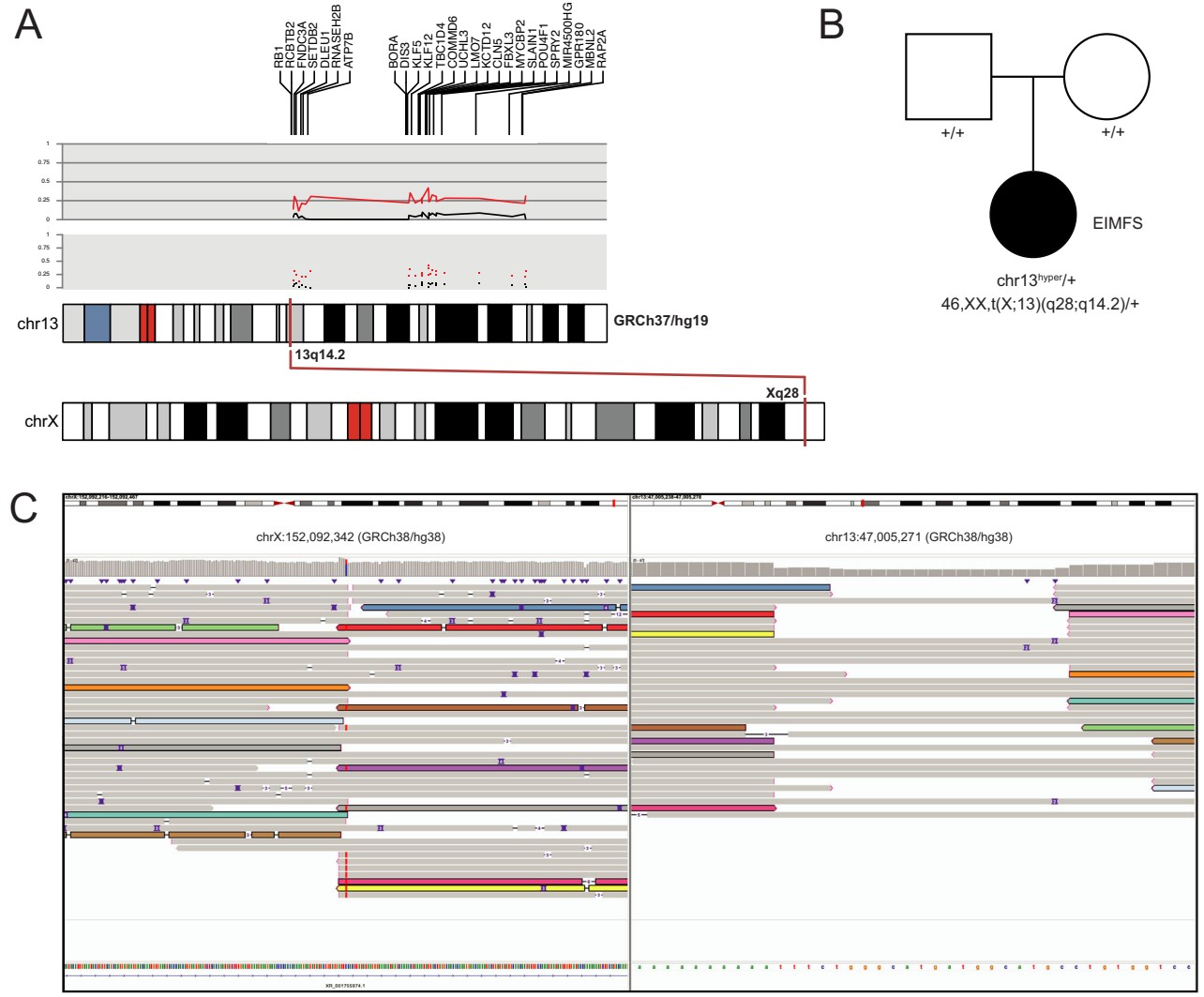

**Fig. 2 | Rare outlier DMR analysis identifies chromosomal hypermethylation caused by X;13 translocation. A** Graphical representation of chr13 rare hypermethylation events in a proband with uDEE. The upper portion of the track displays the genes on chr13 for which hypermethylation events were called. The two grey panels (upper=line, lower=dot) depict β-values for the average of the proband's array replicates (red) and the average of the parents' array data (black) for a representative probe within each DMR (n = 26). Subtle hypermethylation hovering around ~ 25% can be seen for the proband compared to the parents. The lower track shows chromosomal locations of the X;13 translocation. **B** Pedigree showing that chr13 hypermethylation events and the X;13 translocation occurred de novo. **C** IGV view of ONT LRS data for chrX (left) and chr13 (right). Some, but not all, reads spanning the translocation are colored to show that they span the breakpoint. Karyotype plots, as shown in Fig. 2A and others throughout the manuscript, were created using the R package karyoploteR[98].

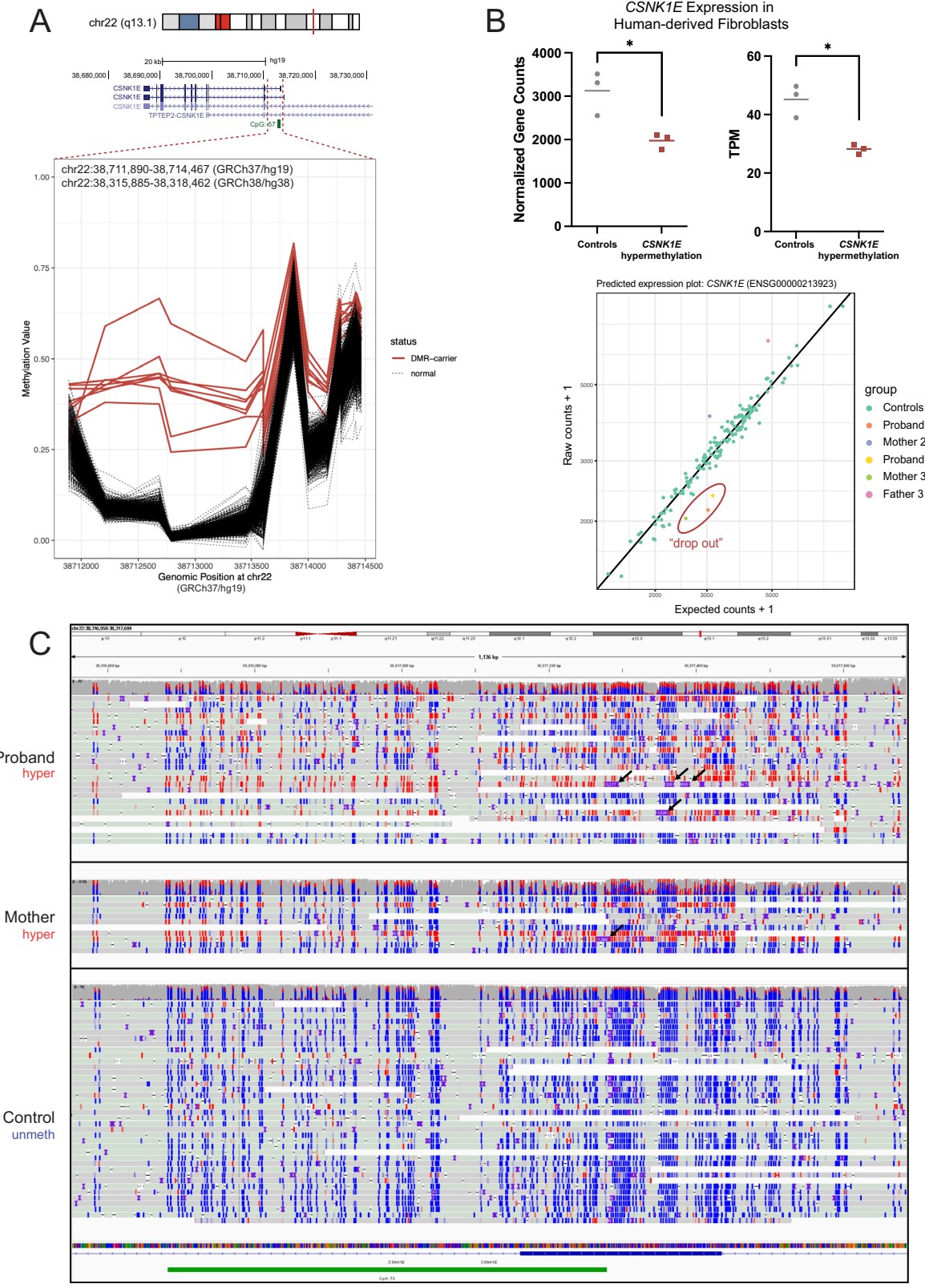

**Fig. 3 | Rare outlier DMR analysis identifies tandem repeat expansions. A** DMR plot depicting outlier hypermethylation of the *CSNK1E* 5′UTR and intron 1 in two probands with uDEEs (three additional replicate samples across both for a total of five samples), one mother, and two unaffected controls (total *n* = 8 red lines) detected through epivariation analysis. **B** The upper panel shows expression values from RNA-seq of human-derived fibroblasts for individuals with *CSNK1E* hyper-methylation (*n* = 3 biological replicates) compared to individuals with control methylation levels (*n* = 3 biological replicates). Significance between groups was determined by a two-tailed paired *t*-test (*p* = 0.029 for gene counts and *p* = 0.0169

for transcripts per million or TPM). **P* < 0.05. A representative predicted expression plot from drop-out analysis using the OUTRIDER algorithm is shown at the lower portion of the panel. See Supplementary Fig. 9 for the individual OUTRIDER plots for each family and significance information. **C** Unphased IGV view of LRS data showing CpG sites that are methylated (red) and unmethylated (blue). The CGG repeat expansion seen in the proband was inherited from the mother (Family 1) and is shown as purple squares denoting insertions in the reads (black arrows); not all reads that are methylated show the insertion as they terminated within the inserted sequence and are clipped by the alignment process.

Supplementary Fig. 9). Thus, we report 3 individuals with uDEEs harboring inherited and de novo *CSNK1E* hypermethylation due to an underlying repeat expansion (*n* = 4 LRS) that leads to approximately 50% reduction in *CSNK1E* expression (*n* = 3 RNA-seq drop-out). No other candidate gene variants for these 3 probands were found by trio GS analysis. However, due to finding this abnormality in seemingly unaffected individuals, one control and one mother (Family 1) in our cohort and others[17], further work is required to determine whether variations in *CSNK1E* cause or contribute to the DEEs.

A male individual with uDEE displayed maternally inherited hypermethylation of the *DIP2B* (MIM:611379) promoter region and exon 1 (Supplementary Fig. 10), due to an underlying CGG-repeat expansion (~1300–2300 bp), previously characterized as fragile site FRA12A[38]. Loss of *DIP2B* is associated with an autosomal dominant neurodevelopmental disorder (NDD) with variable penetrance, including a *DIP2B* repeat expansion in an individual with epilepsy[38].

We detected a rare hypermethylated DMR on the X chromosome in exon 1 of an uncharacterized gene (*BCLAF3/CXorf23*) in a male with uDEE (Supplementary Fig. 10), that was absent in >23,000 unaffected controls (>8000 males)[17]. We validated hypermethylation using targeted EM-seq (Supplementary Fig. 5), and ONT long-read sequencing of the proband and his mother revealed a novel CGG repeat expansion in the proband (~2500–3000 bp, Supplementary Fig. 11) inherited from his mother, who had a smaller expansion (~1700–1900 bp). LRS and standard X-inactivation studies[39] show that the mother has skewed X-inactivation (Supplementary Data 6) of the allele with the expansion, which explains why outlier hypermethylation is not detected from her methylation array data. There are no other candidate variants for the proband's DEE by trio GS. Collectively, these results highlight the detection of repeat-expansion-associated loci based on outlier DMR analysis of DNA methylation array in individuals with uDEEs.

## Rare outlier DMR analysis detects copy number variants

Seven individuals displayed DMRs that were found to be due to underlying CNVs. One individual with uDEE displayed hypermethylation of the promoter region and TSS of *STX1B* (MIM: 601485, Fig. 4A), an established epilepsy gene known to cause generalized epilepsy with febrile seizures plus (GEFs + ). Interestingly, the proband was indeed a member of a family displaying GEFs+ and other epileptic phenotypes (Fig. 4B, Supplementary Phenotype data, Supplementary Fig. 12). We validated the methylation finding in the family members for which blood DNA was available (n = 6 including proband) using targeted EM-seq (Supplementary Fig. 12). Genome sequencing of the proband revealed a 1784 bp deletion encompassing the promoter, the TSS, exon 1, and part of intron 1 of *STX1B*. Importantly, the deletion encompasses the TSS and the first 10 amino acids of the protein encoded in exon 1 (Fig. 4D). We determined the exact breakpoints of the deletion using Sanger sequencing and segregated it among the family members (Fig. 4C, Supplementary Fig. 13). The deletion was confirmed in the proband and present in the affected sister, affected mother, and affected maternal grandmother. The deletion was absent in the unaffected brother and father. Altogether, DNA methylation analysis uncovered a presumably deleterious deletion encompassing an essential portion of *STX1B* gene as a likely pathogenic finding for this family.

One individual with uDEE and one control had a ~10–15 hypomethylated DMRs along chr2 spanning ≥144 Kb (Supplementary Fig. 14A). Short and long-read sequencing analysis revealed this "DMR" was due to a homozygous ~182 Kb deletion encompassing outlier DMRs (Supplementary Fig. 14C). Segregation testing found that the proband inherited the deletion from both parents, who were heterozygous carriers. The CNV was also found on DNA methylation array using the R tool conumee[40] (Supplementary Fig. 14B).

Four individuals with uDEEs and one control had a 686 bp hypomethylated DMR in intron 2 of the gene *LINGO1* (MIM:609791). DNA

methylation array analysis for a proband's mother found that the hypomethylation was at least in part maternally inherited, and short and long-read sequencing revealed that hypomethylation was caused by an underlying ~4 Kb inherited deletion (Supplementary Fig. 15).

Another individual with uDEE had hypermethylation in the 5'UTR of *CFAP36/CCDC104* (Supplementary Fig. 16A, C), which was not present in >23,000 controls[17]. DNA methylation array analysis of both parents indicated it was maternally inherited (Supplementary Fig. 16B), and targeted ONT long-read sequencing revealed a ~500 Kb tandem duplication from chr2:55,034,228-55,536,971 (GRCh38/hg38, Supplementary Fig. 16D). Collectively, these results indicate that outlier DNA methylation can be due to underlying CNVs and that the 850 K methylation array may not have sufficient coverage to detect smaller CNVs. Due to the high population frequencies and inheritance status in the cases of the chr2 deletion, *LINGO1* deletion, and *CFAP36/CCDC104* tandem duplication, we determined they are unlikely to contribute to the individuals' phenotypes. However, these findings illuminate how detected DNA methylation changes are influenced by underlying DNA variation and highlight a novel copy number alteration in *STX1B* as a cause of GEFs+ and other related phenotypes in a family.

## Episignature screening validates pathogenicity of genetic diagnoses and resolves variants of uncertain significance

We next performed episignature analysis, using the EpiSign™ v4 classifier, including 70 conditions associated with 96 genes/genomic regions (Fig. 5). To validate our approach, we included several individuals with causal variants in episignature genes or CNVs and an individual with a VUS. These included sixteen individuals with variants in *CHD2* (*n* = 15 pathogenic, *n* = 1 VUS) and one individual each with a pathogenic variant in *KDM5C*, *SETD1B*, *KMT2A*, or *SMARCA2* (Supplementary Data 2B). We also included two individuals with CNVs, including chr17p11.2 deletion and duplication. Fifteen of the individuals with variants in *CHD2* were positive for the epileptic encephalopathy of childhood (EEOC) episignature[25], also known as the developmental and epileptic encephalopathy 94 (DEE94) episignature. However, one individual with a VUS in *CHD2* was negative for the episignature, and in combination with other clinical evidence the VUS was reclassified as likely benign (Supplementary Phenotype data). The individuals with variants in *KDM5C* (MIM:314690), *SETD1B* (MIM:611055), *KMT2A* (MIM:159555), and *SMARCA2* (MIM:600014) were all positive for the episignatures associated with their disorders. While these individuals were considered solved before episignature screening, the finding was used to support the genetic diagnosis of the individual with a *KDM5C* variant.

Additionally, we identified two individuals with inconclusive results for episignatures despite definitive genetic and clinical findings for the associated syndromes. Inconclusive findings are caused by methylation profiles that partially overlap existing signatures but are not a definitive match. This included an individual with a 17p11.2 deletion inconclusive for the Smith-Magenis syndrome episignature (SMS_del) and a female individual with a 17p11.2 duplication inconclusive for the Potocki-Lupski syndrome episignature (PTLS, Supplementary Fig. 17). In each case, the inconclusive episignature finding is concordant with the genetic diagnosis but yields an inconclusive result potentially attributable to variability introduced by differential CNV breakpoints. Because of this and other factors, inconclusive EpiSign™ results are reported with the caveat that further follow-up or investigation may be warranted if there is a clinical phenotype consistent with the inconclusive episignature in question.

## Episignature screening solves genetically unsolved DEEs

We then tested our cohort of 582 individuals with uDEEs for 70 clinically validated episignatures, leading to a likely diagnosis in five individuals (Table 2). All methylation variant pathogenicity (MVP) scores for episignatures and detailed genomic variant information are in

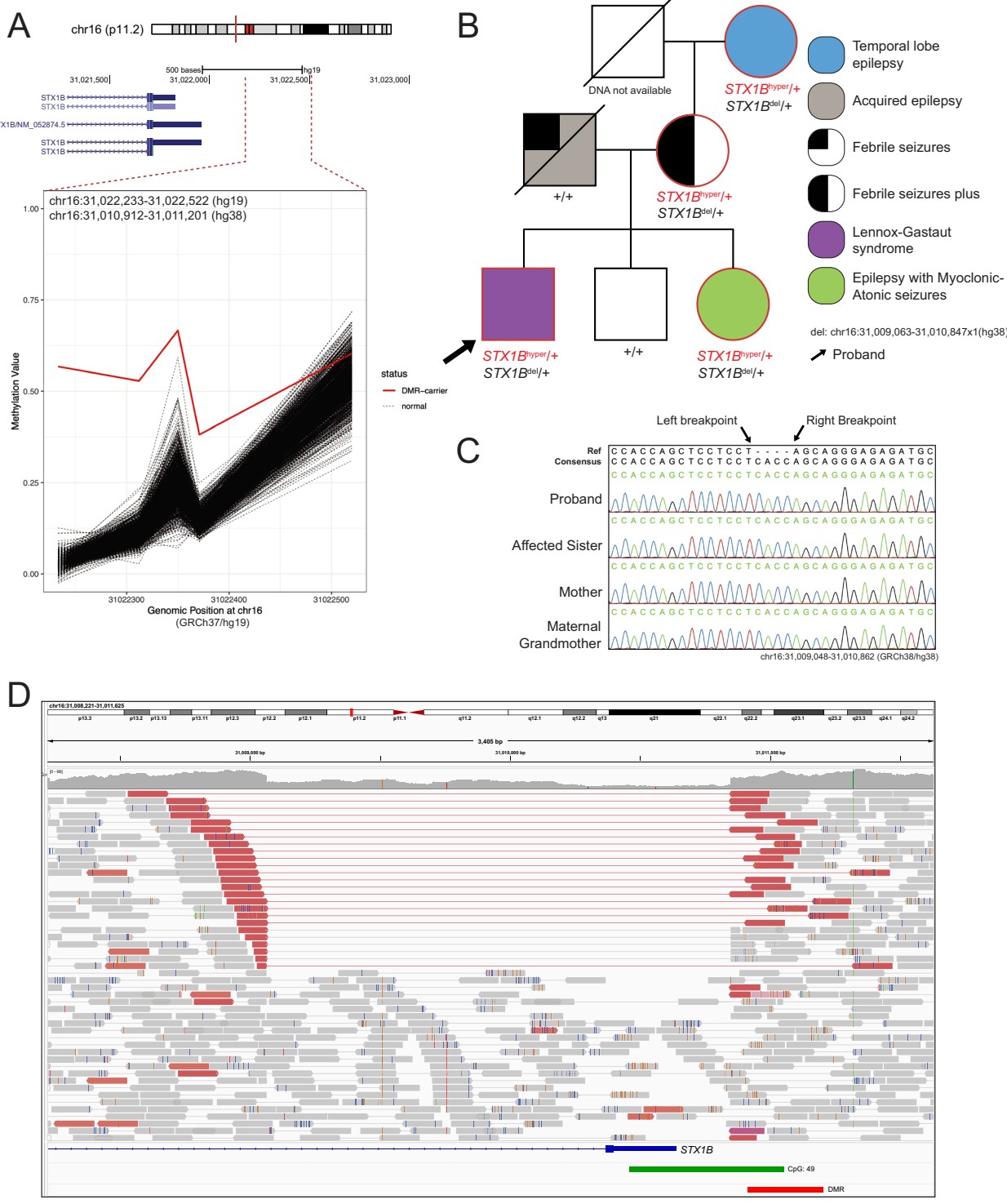

Heterozygous deletion at chr16:31,009,063-31,010,847 (GRCh38/hg38) in proband.

**Fig. 4 | Rare outlier DMR analysis identifies copy number deletion in a family with GEFs + . A** DMR plot depicting outlier hypermethylation of the *STX1B* promoter and TSS in a proband with uDEE detected through epivariation analysis. **B** Pedigree for the immediate family members indicating that the inheritance of the hypermethylation (red) and copy number deletion (detected through genome sequencing of the proband shown in Fig. 3D) resides on the maternal side. The arrow points to the proband. **C** Sanger sequencing validation of the copy number deletion breakpoints in the proband, affected sister, mother, and maternal grandmother. Inserted "CACC" sequence is present between the mapped breakpoints. The father and unaffected brother had no PCR product at the same reaction

conditions, indicating they did not harbor the deletion. Primer pairs with one partner located within the deletion on each side of the breakpoint were used to amplify the wild-type allele as a control (Supplementary Fig. 13). **D** IGV view of genome sequencing data showing a 1784 bp heterozygous deletion encompassing part of intron 1, exon 1, the TSS, and the promoter of *STX1B*. The reads are colored by insertion size and pair orientation and viewed as pairs. The red pairs, which span the breakpoints of the deletion, indicate that the insertion size is greater than expected. The "CACC" insertion sequence is present in the soft clipped bases (not shown).

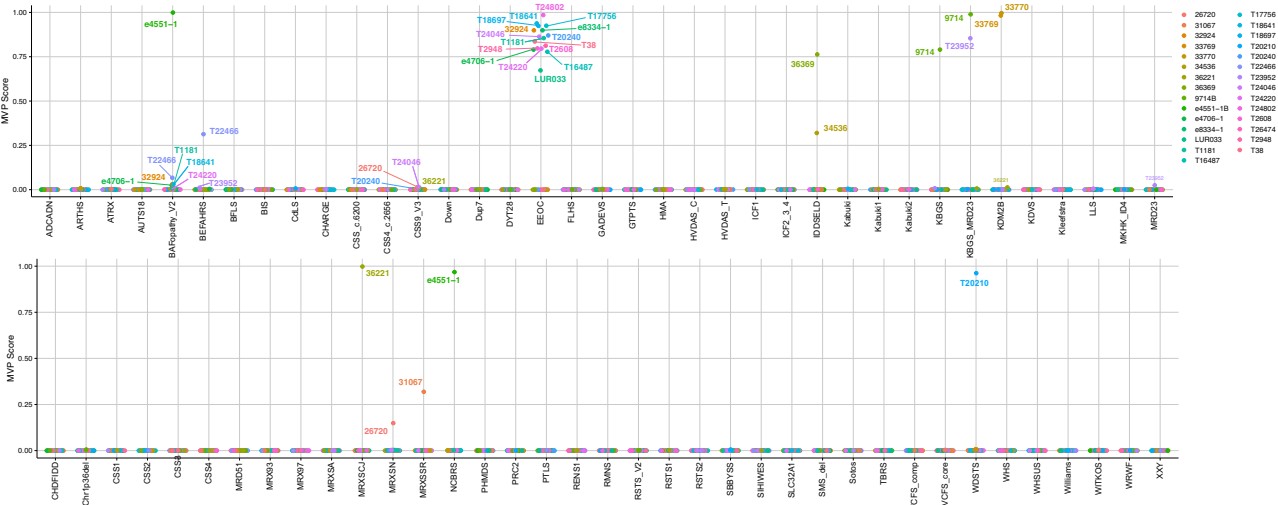

**Fig. 5 | Summary methylation variant pathogenicity (MVP) score for all individuals positive for Episign™ v4 episignature analysis.** A Methylation Variant Pathogenicity (MVP) score (between 0 and 1) was generated to represent the confidence of prediction for the specific episignature on the EpiSign™ v4 clinical classifier that the SVM was trained to detect. Each colored circle represents a different individual and its associated MVP score for each of the episignatures on the EpiSign™ v4 clinical classifier. Final classification for a specific EpiSign™ disorder includes a combination of MVP score, hierarchical clustering, and multi-dimensional scaling (MDS) review.

### Table 2 | Summary of episignatures and causative sequence variants identified in this study

| Gene | Signature | MVP | Genomic Variant (GRCh38/hg38) | Consequence | Inheritance |
|---|---|---|---|---|---|
| *ANKRD11* | KBGS/ MRD23 | 0.854 | chr16:89,284,030 G > A | p.Arg838Ter (p) | de novo |
| | | 0.989 | chr16:89,279,671 C > A | p.Glu2291Ter (p) | de novo |
| *SETD1B* | IDDSELD | 0.763 | chr12:121,822,939 C > T | p.Arg1454Ter (p) | Unknown |
| *TET3* | BEFAHRS | 0.327 | chr2:74,102,031 dup | p.Thr1749HisfsTer5 (p) | Inherited (mat) |
| *UBE2A* | MRXSN | 0.149 | chrX:119,583,172 G > A | p.Ala126Thr (p) | Inherited (mat) |
| *SMS* | MRXSSR | 0.319 | chrX:21,972,570 C > G | p.Arg110Gly (p) | Inherited (mat) |
| *KDM2B* | KDM2B | 0.982 | chr12:121,520,986 C > G | p.Arg349Pro (u) | Inherited (pat) |

List of episignature findings from screening uDEEs. Methylation Variant Pathogenicity (MVP) scores shown. Molecular findings are considered p=pathogenic or u=variant of uncertain significance. Inheritance is listed as de novo, *mat* maternal, *pat* paternal.

Supplementary Data 2C. Two unrelated individuals with uDEEs were positive for the KBG syndrome episignature (KBGS_MRD23) caused by pathogenic variants in *ANKRD11* (Supplementary Figs. 18 and 19). Exome or genome sequencing analysis revealed de novo pathogenic stop-gain variants in both individuals, and phenotypes for each individual are consistent with the diagnosis (Supplementary Phenotype data). One proband had affected siblings and family members (*n* = 8, Supplementary Fig. 19). However, none harbored the *ANKRD11* episignature and neither affected sibling harbored the variant, indicating that there is likely a different explanation for this familial epilepsy. One individual with uDEE was positive for the episignature associated with *SETD1B* (Supplementary Fig. 20). Exome sequencing revealed a pathogenic stop-gain variant in *SETD1B*. Another individual with uDEE harbored the episignature for *TET3* and had a maternally inherited pathogenic stop-gain variant in *TET3* on GS (Supplementary Fig. 21). This remains the likely cause of the individual's DEE as the mother has a milder phenotype including macrocephaly and learning difficulties (Supplementary Phenotype data). One male individual with uDEE was positive for the *UBE2A* episignature (Supplementary Fig. 22). Through exome sequencing, we identified a predicted damaging maternally inherited missense variant absent in gnomAD (c.376 G > A, p.Ala126Thr). Although the variant does not reach likely pathogenic classification using existing ACMG criteria, the prediction scores (REVEL = 0.776, CADD = 26.4, and PolyPhen-2 = 1.00) support pathogenicity; the variant is maternally inherited in an X-linked intellectual disability disorder; and the individual shares multiple phenotypic

features with *UBE2A* disorder. Thus, the variant has been determined to be the most likely genetic cause of disease. Another male individual with uDEE was positive for the episignature for the *SMS* gene on chromosome X (Supplementary Fig. 23). Through ES, we identified a maternally inherited, likely pathogenic missense variant (CADD = 24.2) in the *SMS* gene.

Of the high-confidence episignature findings, only one individual had an established genetic diagnosis in another gene. This individual harbored a de novo variant in *PTEN* with a consistent phenotype of macrocephaly and focal epilepsy but also had the episignature for *KDM2B*. Further analysis identified a paternally inherited missense variant in *KDM2B*. We performed methylation array analysis for the unaffected father and found that he, too, harbored the *KDM2B* episignature. This variant is predicted to be likely pathogenic (LP) by ACMG criteria due to its putative effect on splicing regulation, though assessment of this variant with SpliceAI predicts that it does not have a high likelihood of affecting splicing (Δ score for Donor Gain:0.01). When this criterion is taken away, the designation of LP is reduced to a VUS; other computational predictors assess the impact to be uncertain (REVEL = 0.517). Thus, while it is unlikely that this *KDM2B* variant explains the individual's phenotype, it still represents an underlying DNA change detected through episignature screening, and it remains possible that it has a modifying effect on phenotype. Collectively, we have identified positive episignatures and causal genetic etiologies in five previously unsolved individuals with DEEs through episignature screening.

An additional 40 individuals with DEEs ($n = 32$ unsolved, $n = 8$ solved) and nine controls had inconclusive results for episignatures, consistent with the rate of inconclusive results in previous studies[41]. Of the individuals with DEEs, 4/40 were run across multiple methylation array batches. Three individuals did not reproduce their inconclusive episignature result in the other sample(s). While one individual's inconclusive result did replicate across the different batches, no pathogenic variants were found by GS in the associated gene(s). Of all the individuals with available sequencing data ($n = 27$), none harbored pathogenic variants in the genes associated with episignature findings. While some had overlapping clinical features, most were discordant with the described phenotypes for their inconclusive episignature finding. Additional follow-up will be required to determine whether these inconclusive results are due to array artifacts or have underlying biological or disease-associated meaning. If technical artifacts are ruled out, an inconclusive result may be caused by episignatures in other genes that are yet to be defined and trained against for specificity of the classifier.

### Redefining the CHD2 episignature on the 850 K EPIC array

While episignatures are proven to be clinically useful for diagnosis, little work has been done to investigate how episignatures may inform disease biology by studying DMRs that may impact gene expression. Here, we performed refinement and in-depth analysis of the episignature for the DEE gene *CHD2*. The CHD2 episignature was originally derived using overlapping 450 K and 850 K DNA methylation array probes representing individual CpG sites in $n = 9$ individuals with pathogenic *CHD2* variants[25]. We refer to this signature as the CHD2 450 K episignature (Fig. 6A upper, Supplementary Fig. 24A, Supplementary Data 7). Here, we refine the CHD2 episignature exclusively on 850 K EPIC methylation array probes with data from a cohort of $n = 29$ individuals with pathogenic *CHD2* variants (Fig. 6A lower, Supplementary Fig. 24B, Supplementary Data 7). We refer to this signature as the CHD2 850 K episignature. Of the 200 probes included in the CHD2 850 K episignature, 79/200 are specific to the 850 K EPIC array.

### Comparison of the CHD2 episignature to 55 other clinically validated episignatures

We then compared the CHD2 450 K and 850 K episignatures to 55 other NDD episignatures (57 total including CHD2)[31] by examining shared probes (Fig. 6B, Supplementary Fig. 25), Euclidean clustering (Fig. 6C), probe mean methylation differences (Supplementary Fig. 26), and functional annotations (Supplementary Fig. 27). As expected, the CHD2 850 K episignature shares the most probe overlap with the CHD2 450 K episignature (86/200 or 43%, Fig. 6B, Supplementary Fig. 25). Euclidean clustering was used to examine the relatedness of the episignatures by probe overlap and directionality. The CHD2 850 K episignature shares the closest branchpoint with the MRXSCJ episignature for *KDM5C* of which it shares 7% of its top 500 DMPs. Collectively, both 450 K and 850 K episignatures do not share immediate branches (other than the primary branchpoint) with many other episignatures. This may indicate different sets of predominant pathways underlying CHD2 pathophysiology compared to the other episignatures. Additionally, the CHD2 850 K episignature represents more hypermethylated regions than the CHD2 450 K episignature, as depicted by the mean methylation differences in Fig. 6C and Supplementary Fig. 26. We also performed functional annotation of episignature probes for CpG characteristics and gene regions in relation to the 55 other NDD episignatures (Supplementary Fig. 27). We found that both CHD2 850 K and 450 K DMPs map to predominately the coding regions of genes (46% and 41%, respectively) with a significant difference in the distribution of DMPs in these regions compared with the background probe distribution ($P < 9.06 \times 10^{-69}$ and $P < 2.02 \times 10^{-79}$, respectively). Though the CHD2 850 K episignature represents a higher portion of interCGI (interCpG island) regions compared with the 450 K episignature (43% vs. 31%,

respectively), both are enriched in interCGI regions relative to background probe distribution ($P < 2.26 \times 10^{-121}$ and $P < 9.17 \times 10^{-144}$).

### The CHD2 episignature is associated with differentially methylated regions

Since *CHD2* encodes a chromatin remodeler that has been shown to regulate gene expression[42,43], we investigated whether individual episignature probes are contained within larger DMRs between cases and controls. DMRs could potentially provide a link to downstream gene expression. We first investigated DMRs in an unbiased genome-wide manner by calling DMRs from the 850 K DNA methylation array data ($n = 16$ CHD2, $n = 18$ controls) using bumphunter[44] and DMRcate[45]. We predicted 1684 DMRs from bumphunter and 963 DMRs from DMRcate. These DMRs were intersected, requiring an overlap in the same direction (hyper/hypo) of at least 50 bp, to derive a high-confidence DMR list of 712 overlapping regions (349 hyper, 363 hypo). Representative images of these DMRs are shown in Supplementary Fig. 28. These DMRs directly coincide with 86/200 (43%) CHD2 450 K episignature probes and an increased 90/200 (45%) CHD2 850 K episignature probes (Supplementary Fig. 29, Supplementary Data 8). Thus, the CHD2 episignature is characterized by DMRs, and this overlap increases by four probes for the CHD2 850 K episignature.

### Increased CpG resolution and genomic coverage of differentially methylated regions using whole genome-bisulfite sequencing

Due to limited genomic coverage, DNA methylation arrays can be skewed in their representation of CpGs across the genome, as evidenced by their tendency to bias gene set analyses[46]. To better understand the DMR landscape of CHD2 and investigate DMRs at higher CpG resolution, we performed whole-genome bisulfite sequencing (WGBS) with coverage of > 20,000,000 CpGs on three CHD2 trios and one singleton. We derived 11,019 DMRs from DSS[47], 4078 DMRs from DMRcate[48], and 3655 DMRs that overlap between both callers (2420 hyper and 1235 hypo). To determine the robustness of this approach, we manually inspected DMRs with a methylation difference of at least 20% ($n = 207$ DMRs, 146 hyper, 61 hypo) by examining the reads in all three trios in IGV and confirmed 169/207 DMRs, yielding a true call rate of 81.6%. Representative DMRs called from WGBS are shown in Supplementary Fig. 30. We then investigated the overlap of episignature probes with the WGBS DMRs with a methylation difference of at least 5% and found direct overlap with 76/200 (38%) CHD2 450 K episignature probes and an increased 94/200 (47%) CHD2 850 K episignature probes (Fig. 6D, Supplementary Fig. 29). Thus, considering the increased genomic coverage afforded by WGBS and increased DMRs, it is unsurprising that a higher proportion of CHD2 850 K episignature probes overlap with DMRs (Supplementary Fig. 29, Supplementary Data 8). Notably, for nearly all probes found within DMRs, those DMRs could be better visualized from the WGBS data due to the lack of probe coverage on the array. Thus, we have confirmed using an orthogonal approach with higher CpG coverage that the CHD2 episignature is characterized by DMRs.

We further investigated DMR calls by functionally annotating them using the annotatr[49]. We first examined the representation of CpG islands, CpG shores, CpG shelves, and interCpG Island (interCGI) regions for DMRs (Supplementary Fig. 31). We find that most DMRs called exclusively from WGBS are located at interCGI regions compared to DMRs called from the array or overlap of both, likely due to the bias of gene-enriched regions on the array compared with increased genomic coverage of WGBS. We also annotated DMRs with gene annotations (Supplementary Fig. 32) and found similar patterns across DMRs called by the 850 K array, WGBS, or both, especially for DMRs called with a methylation difference of at least 5% between CHD2 and controls. When compared to three independent sets (Rep-1, Rep-2, Rep-3, Supplementary Data 9) of randomly generated regions of

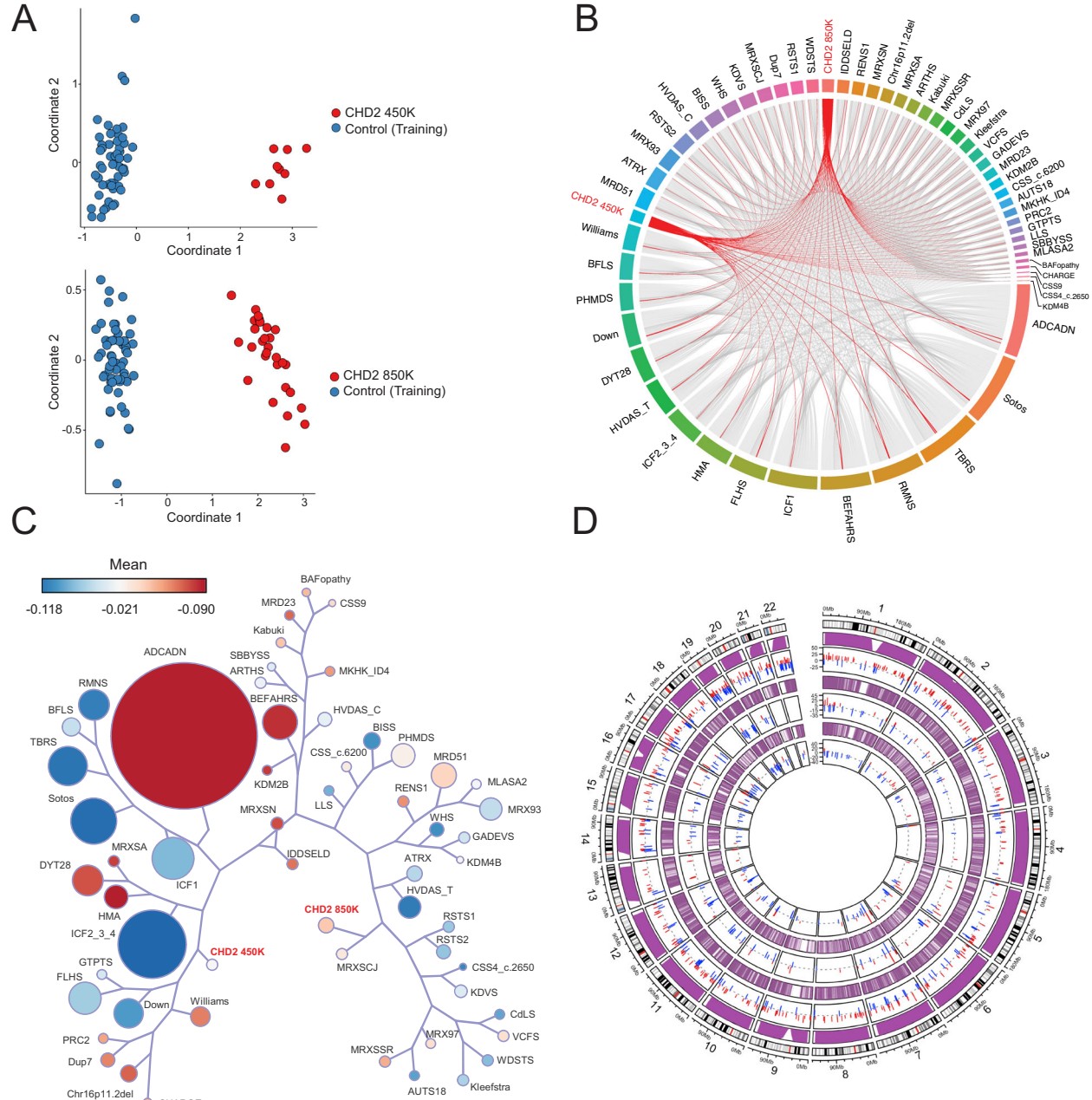

**Fig. 6 | Insights from the CHD2 episignature. A** Multidimensional scaling (MDS) plot showing clustering of individuals with pathogenic *CHD2* variants (red, upper) for the previously described CHD2 450 K (*n* = 9) episignature with shared 450 K and 850 K array probes clusters away from the controls (*n* = 54, blue). The refined CHD2 850 K (*n* = 29) episignature (red, lower) clusters away from unaffected controls (*n* = 58, blue). **B** Circos plot representing shared probes between episignatures. Differentially methylated probes (DMPs) shared between the CHD2 850 K cohort (bold red), CHD2 450 K cohort (red), and 55 other episignatures on EpiSign with functional correlation analysis previously published[31]. The thickness of the connecting lines corresponds to the number of probes shared between the cohorts. **C** Tree and leaf visualization of Euclidean clustering of episignatures. Tree and leaf visualization for all 57 cohorts using the top 500 DMPs for each group (for cohorts with less than 500 DMPs, all DMPS were used). Cohort samples were aggregated using the median value of each probe within a group. A leaf node represents a cohort, with node sizes illustrating relative scales of the number of selected DMPs for the respective cohort, and node colors are indicative of the global mean methylation difference, a gradient of hypomethylation (blue) or hypermethylation (red). **D** Circular karyotype plot showing overlap of CHD2 450 K episignature probes (inner circle, *n* = 200), with CHD2 850 K episignature probes (middle circle, *n* = 200), and WGBS DMRs (*n* = 4 CHD2 vs. *n* = 6 unaffected controls) derived with at least a 15% methylation difference for the condensed visual representation (outer circle, *n* = 411). Each line depicts a probe or DMR where red denotes hypermethylation and blue denotes hypomethylation. The purple tracks depict coverage of the 450 K array probes (inner), 850 K EPIC array probes (middle), and WGBS reads (outer). Refer to Supplementary Fig. 33 for linear karyotype DMR plots for chr1-22.

comparable number (*n* = 4767) and length (*n* = 50-3100 bp) representing background, the combined CHD2 episignature probes and DMRs (*n* = 4767) are enriched in gene regulatory regions (enhancers, promoters, and bivalent regions), transcription factor binding sites (TFBS), and DNase sites (Fig. 7B, C, D, Supplementary Data 10).

Although CHD2 episignature and DMR insights are limited to the blood in our study, this work supports further investigations into CHD2 methylation of brain-relevant tissue types, such as cultured neurons, brain organoids or, when available, post-mortem tissue. Notably, we show how the global CHD2 episignature is characterized by DMRs

(Fig. 7A, Supplementary Fig. 33) enriched in functional regions, and therefore, poised to affect underlying disease biology.

## Discussion

A major challenge in rare disease genetics is determining molecular causes in unsolved cases. Even if ES or comprehensive GS of trios identifies all de novo and recessively inherited coding and noncoding variants, prioritizing and functionally interpreting candidate variants is challenging. In the case of the DEEs, this difficulty is further compounded by immense phenotypic and genetic heterogeneity. Genome-wide DNA methylation analysis represents an innovative approach to discovering genetic etiologies by investigating rare DMRs and screening for DNA methylation signatures. Notably, rare DMRs and episignatures can be assessed with cost-effective, high-throughput DNA methylation arrays using blood-derived DNA. Here, we performed genome-wide DNA methylation analysis on 582 individuals with uDEEs and identified causal or candidate etiologies in 12 individuals: six from rare DMR analysis (Table 1) and six from episignature screening (Table 2). Thus, the diagnostic yield of genome-wide methylation analysis in individuals with uDEEs is 2%, similar to the added diagnostic yield of GS after ES or gene panel[50,51]. A study of unsolved ND-CA showed a similar 2-3% increase in diagnostic yield using episignature analysis[52].

We have performed rare outlier DMR analysis of methylation array data for a cohort of individuals with uDEEs and uncovered various underlying DNA variants using ONT long-read sequencing. These include a X;13 translocation, CGG repeat expansions, and copy number variants. We first validated a subset of outlier DMRs using targeted EM-seq enriched for 3.98 M CpGs, a highly effective bisulfite-free, enzyme-based conversion method for detecting CpG methylation by sequencing. Targeted EM-seq has several advantages to bisulfite-based array approaches, including minimizing DNA damage, lowering input requirements (picograms of DNA), and detecting more CpGs[53]. We found that all DMRs were confirmed using the EM-seq approach, and the greater number of CpGs detected compared to the methylation array afforded higher resolution to interpret DMRs. Future high-throughput DNA methylation analyses could consider using EM-seq for validation or discovery.

We report an individual with 26 outlier hypermethylation events along chr13q detected through the rare DMR analysis. Using ONT whole-genome long-read sequencing, we identified a de novo X;13 translocation showing that the hypermethylation identified the likely cause of disease. This discovery was enabled without the need for live cellular material, which is typically required by classical cytogenetics approaches. This child passed away at 7-months-old due to the severity of the disease, and this approach provided a diagnosis postmortem using banked genomic material.

We also found that several individuals displayed hypermethylation of loci associated with known or novel CG-rich repeat expansions. These regions include the 5'UTR and intron 1 of the epilepsy candidate gene *CSNK1E*, the 5'UTR of the neurodevelopmental disorder gene *DIP2B*, and the 5'UTR of the uncharacterized gene *BCLAF3*. We report the occurrence of hypermethylation, a CGG repeat expansion, and reduced expression of *CSNK1E* among three unrelated individuals with uDEEs and a mildly affected mother. *CSNK1E* has been implicated in the circadian rhythm[54,55], and variation causes a familial advanced sleep phase syndrome (FASPS)[56]. Variation also produces a rapid eye movement phenotype in a knockout mouse model[57]. Interestingly, all our probands with DEEs and the mildly affected mother with *CSNK1E* hypermethylation and a repeat expansion report sleep-related phenotypes (Supplementary Phenotype data). Our results indicate that there is an enrichment of *CSNK1E* hypermethylation in individuals with DEE compared to controls in our cohort combined with those previously reported[17] (Fisher's Exact $P = 0.0276$), suggesting that further studies to determine if *CSNK1E* variation contributes to DEEs are warranted.

One male proband with uDEE displayed de novo outlier hyper-methylation in a region annotated as intergenic on the GRCh37/hg19 genome build and at the 5'UTR of *BCLAF3* on the GRCh38/hg38 genome build. Using ONT long-read sequencing, we discovered a novel CGG repeat expansion in exon 1 of *BCLAF3* in this proband inherited from his unaffected mother. The mother's long-read data displayed skewed X-inactivation against the expanded allele. Skewed X-inactivation may explain why the mother does not have a detectable DNA methylation abnormality at this locus and could provide a mechanism for her to circumvent any functional consequences of the *BCLAF3* abnormality. While *BCLAF3* has been previously predicted to be a potential disorder-associated gene on chrX[58], little is known about its function or disease associations. Thus, further work is needed to investigate whether this abnormality is present in other individuals and if loss of this gene on chrX in males could cause a DEE.

Seven individuals displayed DMRs due to underlying CNVs, one of which we found to be likely pathogenic. Hypermethylation of the *STX1B* TSS and promoter from a proband with uDEE revealed a 1784 bp het-erozygous deletion in GS - 65 bp away, which was confirmed to be present in an affected sister, affected mother, and affected grand-mother. This deletion encompasses the promoter region, the TSS, exon 1, and part of intron 1 of *STX1B*, resulting in probable loss of function. Importantly, the deletion is unlikely to be detected through standard microarray approaches due to its small size and may escape gene panels and exome sequencing, which would not detect the non-coding portions. DNA methylation served as a signpost of the cause of this family's epilepsy and led to the identification of a pathogenic variant.

We performed episignature screening of our uDEE cohort using the EpiSign™ v4 classifier, which contains 90 episignatures representing 70 disorders encompassing 96 genes/genomic regions. We found seven individuals with uDEEs harbored positive episignatures concordant with their phenotypes. We reviewed or reanalyzed available or newly generated ES or GS data and identified pathogenic variants in the episignature-associated genes in 6/7 individuals. In the individual with a pathogenic *SETD1B* variant, the father was unavailable for genetic testing to segregate the sequence variant. Thus, the positive episignature finding provided supportive information for genetic diagnosis in lieu of inheritance data. Episignatures can serve to screen for disorders that have broad, overlapping phenotypes and identify individuals who may not have the classical features of specific neuro-developmental syndromes or DEEs. For instance, most DEEs have a phenotypic spectrum, so individuals with different etiology, develop-mental trajectories, or subtle dysmorphic features may escape diag-nosis until a molecular etiology is found.

The top 27 most implicated genetic causes of DEEs explain 80% of DEEs[7]. However, only 1/27 genes (*CHD2*) has a clinically validated epi-signature. Like *CHD2*, 58/59 genes with robust episignatures localize to the nucleus and are associated with DNA binding, transcriptional regulation, and histone interactions. Since DNA methylation occurs in the nucleus, most genes for which episignatures have been derived are directly or indirectly involved in the epigenetic and transcriptional machinery. Whereas the top 27 DEE genes are associated with a range of cellular processes[5], only a minority are associated with direct DNA interactions, and only 10 of the top 27 most frequent DEE genes are annotated to localize to the nucleus at least partially. The only gene with a clinically validated episignature not involved in any nuclear activity is *SLC32A1*, which encodes solute carrier family 32 member 1 (*SLC32A1*, MIM:616440) responsible for inhibitory neurotransmission, and variants in this gene cause a DEE[59]. Unfortunately, *SLC32A1* is not among the most common ~60 DEE genes. Therefore, the diagnostic utility of episignatures for DEEs would increase when we can con-fidently derive episignatures for more DEE genes, such as ion channel, synaptic transmission, and metabolic genes.

Episignature derivation is further complicated by the existence of variant-specific episignatures that exist for a subgroup of variants

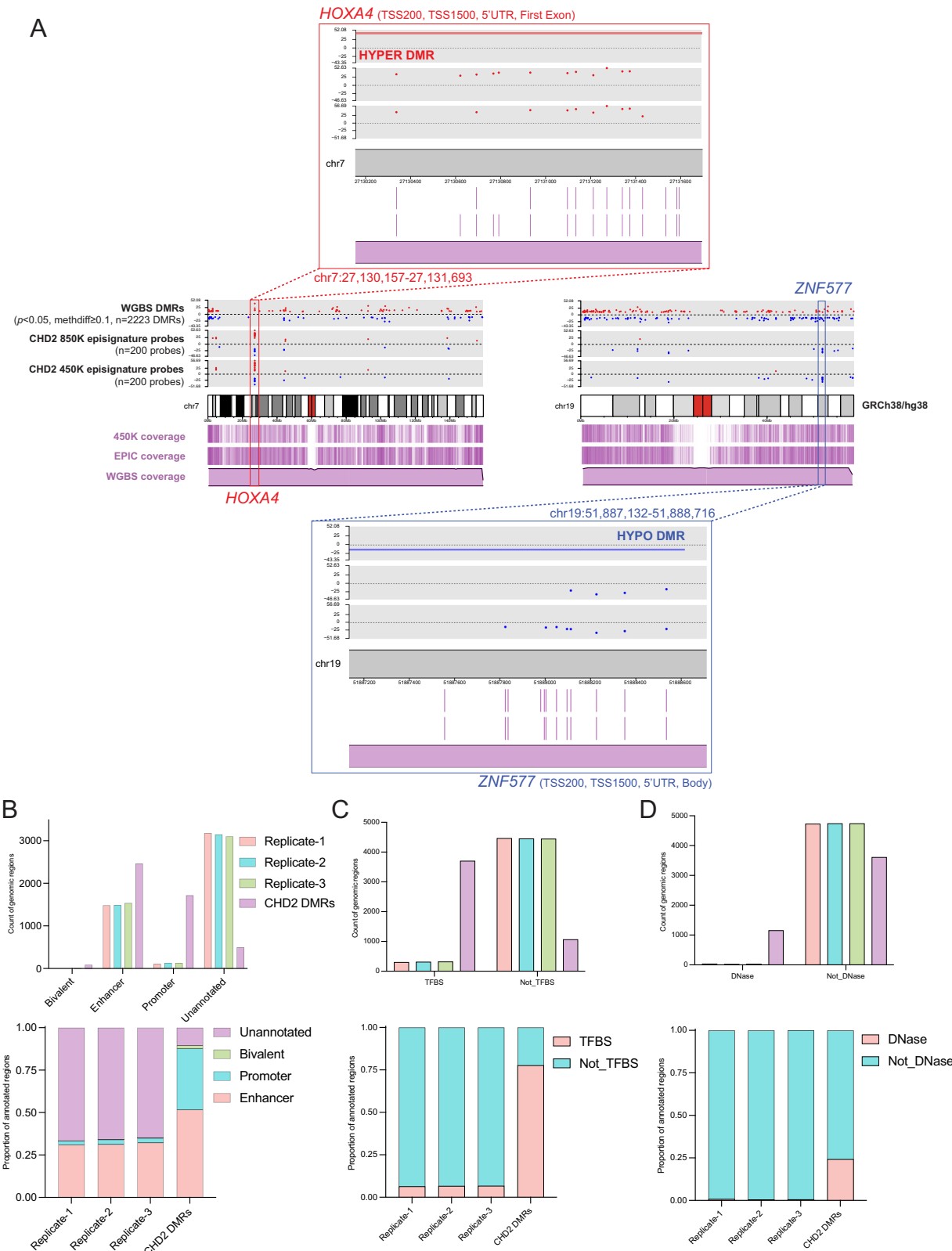

within a gene (e.g. *SMARCA2*[29,60]) or a set of common genes within similar pathways (e.g. Coffin-Siris syndrome episignature, due to variants in *ARID1A* (MIM: 603024), *ARID1B* (MIM:614556), *SMARCB1* (MIM:601607), and *SMARCA4* (MIM:603254), and *SOX11* (MIM:600898)[60]. Thus, there is not only a need to derive episignatures for more epilepsy-related genes but also to analyze variants for testing based on variant type (i.e. missense, nonsense) and protein domain,

which may segregate with phenotypes. For instance, our cohort included two females with solved DEEs and pathogenic truncating variants in the *SMC1A* gene located on chromosome X. Neither had a positive episignature for *SMC1A* for Cornelia de Lange syndrome (CdLS), which is usually due to missense or in-frame small indels proposed to have a dominant negative effect. Truncating, loss-of-function variants, however, are found exclusively in girls with DEEs.

**Fig. 7 | The CHD2 Episignature is associated with DMRs enriched in regulatory regions. A** Karyotype plots depict direct overlap of multiple CHD2 episignature probes with DMRs (left=hypermethylated region, right=hypomethylated region) called from WGBS (*n* = 4 CHD2 vs. *n* = 6 unaffected controls). For each karyotype plot, the three grey tracks (upper panel above chromosome) depict individual red (hyper) or blue (hypo) dots for WGBS DMRs (upper), CHD2 850 K episignature probes (middle) and CHD2 450 K episignature probes (lower). The scale denotes the methylation difference between CHD2 relative to controls. Three purple tracks (lower panel below chromosome) depict the coverage for the 450 K array probes (lines, upper), 850 K array probes (lines, middle), and WGBS reads (distribution, lower). The coverage track for the WGBS was taken from a representative sample after inspecting the average coverage values across all the samples. "Zoomed" in karyotype plots of the boxed regions of probes clustering around for *HOXA4* (hyper) and *ZNF577* (hypo) are shown above or below the karyotype plots. Gene annotations are noted as within 200 bp or 1,500 bp of the transcription start site (TSS200, TSS1500), the 5' untranslated region (5'UTR), or gene body (Body). For both examples, multiple episignature probes map to DMRs. **B** Enrichment (values in Supplementary Data 10) of CHD2 episignature probes and DMRs (*n* = 4767 from array and WGBS, Supplementary Data 8) in various regulatory regions (bivalent regions, enhancers, and promoters) annotated with GREEN-DB[99] compared to randomly generated genomic regions (Replicate-1, Replicate-2, and Replicate-3, Supplementary Data 9) of equal number (*n* = 4767 regions each) and varying, comparable sizes (50–3100 bp in length). Fisher's Exact *P* values were calculated using two-sided Fisher's Exact tests to determine the significance of enrichment. In all cases of CHD2 vs. Replicate-X, Fisher *P* < 2.2e⁻¹⁶. Counts of the regions annotated displaying enrichment in CHD2 are shown in the upper bar plot, and relative proportions are shown in the lower plot. Transcription factor binding sites (TFBS) annotation counts are plotted in **C**. (Fisher *P* < 2.2e⁻¹⁶). DNase sites annotation counts are plotted in **D**. (Fisher *P* < 2.2e⁻¹⁶).

The difference in underlying disease mechanism likely impacts the composition of the distinct probe sets contained within the episignatures. Discordant or unusual findings like this example underscore additional considerations when deriving and interpreting episignatures. We came across five individuals reported as male whose methylation pattern on the X chromosome suggested two X chromosomes. Of 2/5 of these individuals who had LRS, a genotype of XXY was confirmed, which is consistent with a diagnosis of Klinefelter syndrome. More unexpected and incidental findings will arise as a greater number of episignatures are derived, and methylation testing becomes more routine.

Episignatures for many epilepsy-related genes are currently in development. As more episignatures are clinically validated, re-analysis of previously generated methylation array data from unsolved individuals will identify pathogenic findings, akin to re-analysis of exome sequencing data for new epilepsy genes years after initial sequencing was performed[61]. We found that episignature analysis was useful for clarifying VUSs, including an individual annotated as solved for *CHD2* displaying a VUS, which was re-assessed as benign based on a negative CHD2 episignature result. We anticipate that episignatures will also be useful for interpreting the impact of noncoding variants.

There are additional considerations when determining the utility of DNA methylation analysis for the molecular diagnosis of individuals with DEEs. Firstly, the diagnostic utility will vary depending on when the individual receives the test relative to other genetic testing modalities. In our study, we analyzed DNA from individuals with DEEs who had remained unsolved after undergoing extensive genetic testing, including gene panels, microarrays, exome, and genome sequencing. As DNA methylation testing becomes increasingly accessible to newly diagnosed individuals with DEEs and as the number of epilepsy-relevant genes with robust episignatures grows, the utility of DNA methylation analysis in uDEEs may increase and guide which regions should be sequenced to identify causal variants.

DNA methylation information can be readily assessed from both ONT long-read sequencing and PacBio long-read sequencing data. Therefore, when long-read sequencing becomes more available, there is potential for an "all-in-one" approach to genetic testing whereby individuals can simultaneously be assessed for sequence variants, structural abnormalities, and rare DNA methylation changes. While it is advantageous to study rare DMRs and their potential underlying DNA defects using the same technology, applying episignatures to long-read sequencing data is uncertain and may require new computational approaches to re-derive and validate episignatures on each platform. As long-read sequencing produces far more data than arrays (>20,000,000 CpGs versus ~850,000 CpGs), this will offer an opportunity to interrogate DNA methylation more broadly and deeply.

As advances in sequencing technologies allow DNA methylation datasets to get larger, there will be a need to analyze comparative data

from controls to generate population-level reference information. For our DMR analysis, we leveraged 450 K DNA methylation array outlier DMR calls generated from peripheral blood-derived DNA for > 23,000 control individuals[17]. Where possible, we used these data to approximate population frequencies for the DMRs we derived. However, this reference information is not available for 850 K exclusive DMRs or whole-genome sequencing DMRs. Thus, interpreting DNA methylation data for uDEEs and other unsolved genetic disorders will improve as we understand more of the methylome, including regions that were only recently resolved on the T2T genome build[62], using appropriate reference datasets from diverse populations.

While episignatures provide a robust readout of the genetic etiology, they are composed of individual array probes representing singular CpG sites that might not contribute to understanding the underlying disease mechanism. Given that *CHD2* is the most frequent DEE gene with a robust episignature and has a biological role as a chromatin remodeler, we were interested to use the episignature to understand how DNA methylation relates to underlying CHD2 pathophysiology. First, we re-defined the episignature on exclusively 850 K array probes with an increased sample size from *n* = 9 to *n* = 29 individuals with *CHD2* pathogenic variants. Using DNA methylation array and WGBS, we show that the CHD2 episignature is associated with DMRs between cases and controls. In a recent study, investigators derived DMRs for individuals with pathogenic *HNRNPU* (MIM:617391) variants versus controls in methylation array data from peripheral blood-derived DNA and reported 19 DMRs called with DMRcate (Fisher *P* < 0.01, betacutoff = 0.05, minCpG = 5)[63]. The comparative number of DMRs we derived for CHD2 versus control methylation array data under the same conditions using DMRcate is 474 DMRs. This increased number of DMRs may represent the inherent function of CHD2 as a chromatin remodeler that interacts directly with the DNA, whereas HNRNPU forms complexes with RNA. Furthermore, a subset of CHD2 episignature probes overlap with DMRs in the TSS/5'UTR of developmentally relevant genes and might regulate expression (Supplementary Data 8). For instance, a cluster of hypermethylated episignature probes for the CHD2 450 K and 850 K episignatures are contained within a larger hypermethylated DMR in the TSS and 5'UTR of *HOXA4* (Fig. 7A). However, *HOXA4* is not expressed in the blood, and, therefore, would not be expected to be impacted by differential methylation. Thus, we have shown that CHD2 is associated with DMRs in the blood that correspond with the episignature and are enriched in functional regions (enhancers, promoters, bivalent regions, TFBS, and DNase sites). Our work suggests that future studies should investigate the CHD2 episignature in disease-relevant tissue types where DMRs are likely to contribute directly to gene dysregulation and disease pathogenesis.

Here, we have utilized various DNA methylation analyses to identify causative and candidate etiologies in 2% of our cohort of 582 individuals with uDEEs. While DNA methylation does not explain the

majority of DEEs, methylation array yield is comparable to the current added utility of GS[50,51] and remains a low-cost approach that can detect missed genetic etiologies and propose new molecular candidates. Importantly, this yield is expected to increase over time as we interrogate the functional consequences of rare DMRs and better understand which genes and pathways exhibit episignatures, including unraveling inconclusive episignature results. We have also investigated the episignature for the DEE gene *CHD2* in-depth and have provided evidence that the CHD2 episignature is associated with DMRs. DMRs are enriched in functional regions and may affect gene expression, especially in disease-relevant tissue types. Furthermore, CHD2 episignatures and associated DMRs may have potential as a biomarker readout for therapeutic testing, as the DNA methylation might potentially be reversed with targeted treatment. Thus, our work highlights the impact of investigating DNA methylation in DEEs, both for the genetic diagnosis of unsolved cases and to augment our understanding of underlying disease function toward the future development of targeted therapies.

## Methods

### Cohorts

Our cohorts consist of 593 affected individuals (43% female) with uDEEs and 475 healthy controls (47% female) (Fig. 1B, Supplementary Data 1A, Supplementary Methods). An additional 148 analytical controls (60% female) were included for validation. Individuals with DEEs were recruited from investigators' research and clinical programs[64,65]. Methylation array data for healthy controls were drawn from a public database[66] ($n = 111$), an internal institutional database (SJLIFE, $n = 335$), and unaffected parents or siblings ($n = 29$) of probands with DEEs (Supplementary Methods). Eight family members with epilepsy were studied to identify familial methylation patterns (shared rare DMRs or episignatures). Analytical controls, including i) six individuals each with a disease-associated rare DMR, ii) 26 individuals with a pathogenic variant in a gene or CNV associated with an episignature, and iii) 116 individuals with a pathogenic variant in a gene without a known episignature, were used to validate positive and negative rare DMR and episignature findings in the DEE cohort. After quality control and normalization (described below), there were 582 remaining individuals with uDEEs (43% female) who had undergone extensive molecular testing: 79% (458 individuals) had a gene panel, 51% (298 individuals) microarray or karyotype analysis, 75% (435 individuals) ES, and 40% (232 individuals) GS. Collectively, 97% (562 individuals) had at least one sequence-based investigation (gene panel, ES, or GS). There were also 461 healthy controls (47% female), 143 analytical controls (57% female), and eight affected family members for DNA methylation analysis. This study was approved by the Institutional Review Board (IRB) of St. Jude Children's Research Hospital (SJCRH). Written informed consent was provided by parents or legal guardians of individuals with DEEs with local IRB approval from SJCRH, Austin Health (Australia), the University of Washington (UW), and the National Institutes of Health (NIH). For any photographs shown in the supplement, we affirm that the patients and representatives have consented to open-access publication and have seen the photos in the context of the publication.

### Methylation array

All data were from peripheral blood-derived DNA, except for five analytical control samples used for outlier DMR analysis: saliva-derived DNA from one female individual with BSS and her mother (carrier) and lymphoblastoid cell line (LCL)-derived DNA from three individuals, including two males and one female, with Fragile X syndrome (Coriell). These samples were used as positive controls to validate the outlier analysis, and then removed from the final analysis to minimize potential cell type differences. DNA was extracted from peripheral blood samples using standard protocols, with

approximately 250–500 ng of DNA bisulfite converted. The Illumina Infinium MethylationEPIC v1.0 (850 K array) bead chip arrays (processed according to the manufacturer's protocol) interrogate > 850,000 individual CpG sites, including CpG islands, promoter regions, gene bodies, FANTOM5 enhancers, and proximal ENCODE regulatory elements[67].

Of 1224 individuals included, three individuals were run in triplicate, and 29 were run in duplicate to produce a total of 1259 blood-derived DNA methylation array samples before quality control and processing. Each sample consisted of data for > 850,000 probes that were rigorously quality-controlled for the removal of outlier samples as opposed to outlier regions of interest. All data were combined and loaded into the R package minfi[68] for quality control and normalization. Samples judged to be of poor quality (> 1% of probes that failed) and samples that were deemed outliers based on manual inspection of the principal component analysis (PC1 and PC2), using β values for probes located on chromosome (chr) 1, were removed (Supplementary Fig. 1). Individual CpG probes that failed (detection $p > 0.01$) in > 10% of samples were removed; also, probes overlapping with common SNPs and those previously reported as cross-reactive were removed[67,69]. Since samples were run in multiple batches and at different institutions, we visually examined the PCA plot for batch effects. The only batch effect observed was on PC1 between the SJLIFE unaffected control cohort and the rest of the samples analyzed (including both cases and controls). We used the R package SVA[70] for batch correction using the ComBat method and confirmed the elimination of the batch effect (Supplementary Fig. 1, Supplementary Data 1B)[71]. We estimated blood cell type composition for six cell types (CD8T, CD4T, NK, B-cell, monocytes, and granulocytes) from β values for each sample[72]. Samples containing outlier cellular fractions defined as ≥ 99th percentile + 2% or ≤ 1st percentile − 2% for at least two of the six cell types were also removed. Methylation array intensity values on the sex chromosomes (X, Y) were used to infer the sample sex and compared to the clinically reported sex. Samples with sex mismatches were removed. Samples were separated into inferred sex (males and females) for all downstream analyses of sex chromosomes. Quality control and filtering left 1226 samples across 1194 individuals (26 individuals in duplicate and three individuals in triplicate across batches) assayed by the 850 K array and 793,009 probes (775,431 autosomal probes and 17,578 sex chromosome probes) (Supplementary Data 1C).

### Identification and annotation of rare epivariants

To identify outlier DMRs, we used a sliding window approach as previously described[17]. In brief, this algorithm employs user-defined quantile thresholds to determine outlier β values across multiple CpG sites. Per 1 Kb window, at least three consecutive CpG sites must exhibit outlier β values in the same direction (hyper or hypo) for a sample compared to the rest of the cohort to be considered an outlier DMR. We considered β values above the 99.25th percentile plus 0.15 as hypermethylated, and those below the 0.75th percentile minus 0.15 as hypomethylated for analysis of the autosomes (chr1-chr22). Since samples were split into inferred sex (males and females) for analysis of the sex chromosomes, the stringency was adjusted accordingly to 99th plus 0.15 for hypermethylated and 1st percentile minus 0.15 for hypomethylated. Samples with over 100 rare DMRs on the autosomes during the initial analysis ($n = 7$) were removed from the final analysis as this is thought to be artifactual and may interfere with real signal. DMRs were then annotated to inform functional interpretation using HOMER[73] and including overlap with UCSC RefSeq gene bodies and promoter regions, defined as ± 2 Kb of the transcription start sites (TSS), known CpG islands (CGIs), repetitive-element information (RepeatMasker and SimpleRepeats), imprinting control centers[74], CTCF-binding sites[75], gene molecular function information[73], OMIM phenotype[76], average brain expression using bulk RNA-seq data from

the GTEx Portal, and in-house epilepsy- and candidate-gene lists to prioritize candidates but not as exclusion criteria. Additionally, a recent study delineated the rare DMR landscape in the human population by examining 450 K methylation array data from > 23,000 individuals[17]. Regions from those data were checked against our DMRs where possible to determine the frequency at which each DMR occurs in the population. Based on this annotation information, DMRs were prioritized by four features: (1) a low or negligible population frequency; (2) a well-annotated genomic location, such as in or near known epilepsy and candidate genes; (3) recurrence in multiple individuals; and (4) manual inspection of DMRs, including flanking regions.

### Development of a DNA methylation array analysis and visualization pipeline

We developed MethylMiner, a methylation array analysis pipeline tailored toward discovering rare epivariants with interactive data visualization. The pipeline requires standard input files, raw signal.idat files containing each sample's green and red channels, and a metadata sheet including sample names, sentrix IDs, reported sample sex, and sample group (if applicable). In brief, the pipeline performs quality control and normalization as described to derive output files, including quality control reports, β values, M-values, and bigWig files for quick and convenient visualization in the integrative genomics viewer (IGV)[77]. The pipeline then performs the outlier DMR analysis (using scripts derived from the GitHub repository: https://github.com/AndyMSSMLab/Methylation_script) based on user-defined quantile thresholds and outputs the DMRs and annotations into a tabulated sheet. This annotated list of DMRs is then used as input for the interactive data visualization in JupyterDash, which allows users to interact with plots for quality control metrics, DMR annotations, and DMR genomic tracks. Static DMR plots, like those displayed throughout this manuscript, were created using the <AndyMSSMLab/Methylation_script/blob/main/plotDMR.R> script. The MethylMiner pipeline is hosted on our GitHub page (https://github.com/stjude-biohackathon/MethylMiner).

### Validation of outlier DMRs using enzymatic methyl-sequencing

We performed targeted Enzymatic Methyl-sequencing (targeted EM-seq) enriched with the Twist Human methylome panel targeting 3.98 M CpGs through 123 Mb of genomic content. Targeted EM-seq of peripheral blood-derived DNA was used to validate a subset of outlier DMRs, including $n = 2$ positive control DMRs (*XYLT1* and *FMR1*) and $n = 29$ DMRs-of-interest called amongst $n = 6$ individuals with uDEEs and $n = 4$ family members. EM-seq library preparation, target enrichment, and sequencing were performed using standard protocols[53]. Reads were processed using the "nf-core/methyseq" pipeline with the '--emseq' flag. For detailed EM-seq methods, please refer to Supplementary Methods.

### Identification of structural variants with long-read sequencing

We used both targeted and whole-genome LRS on the ONT platform to validate rare DMRs and identify candidate disease-causing variants at or near the site of interest (Supplementary Data 2A). Targeted LRS using the "read-until" function was performed on an ONT GridION using a single R9.4.1 flowcell as described previously[78]. At least 100 Kb of sequence was added to either side of the target region for capture. Libraries for GS were prepared using the ligation sequencing kit (SQK-LSK110) following the manufacturer's instructions, then loaded onto a single flowcell (FLO-PRO110, R9.4.1) on a PromethION and run for 72 h with one wash and reload. All data were base called using Guppy 6.3.2 (ONT) with the superior model including 5mC methylation. Reads were aligned to GRCh38/hg38 using minimap2[79], SNP and indel variants were called using Clair3[80], structural variants were called using Sniffles[81], SVIM[82], and CuteSV[83], and phasing was performed using LongPhase[84]. Aligned and phased bam files were visualized in IGV[77].

### Episignature testing

Data were blinded and submitted to the clinical bioinformatics laboratory [Molecular Diagnostics Laboratory, London Health Sciences Centre (LHSC), Western University, London, Canada] through a secure file transfer protocol and stored on encrypted servers. The data analysis pipeline was adapted from previously described methods[25] as summarized in Fig. 1A. Importantly, probes with a detection *p*-value > 0.01, probes located on the X and Y chromosomes, probes that contained SNPs at the CpG interrogation or single-nucleotide extension sites, and probes that are known to cross-react with other genomic locations were removed[67,69]. DNA methylation data for each sample were compared to clinically validated DNA methylation signatures for all disorders which are part of the EpiSign™ v4 clinical test[85]. The reference database EpiSign™ Knowledge Database (EKD) includes thousands of clinical, peripheral blood DNA methylation profiles from disorder-specific reference and normal controls (general population samples of various ages and racial backgrounds). Individual DNA methylation data for each individual were compared with the EKD using the support vector machine (SVM) based classification algorithm for EpiSign™ disorders. A Methylation Variant Pathogenicity (MVP) score between 0 and 1 was generated to represent the confidence of prediction for the specific disorder the SVM was trained to detect. Conversion of SVM decision values to these scores was carried out according to the Platt scaling method[86].

Classification for a specific EpiSign™ disorder included a combination of MVP score, hierarchical clustering, multidimensional scaling (MDS) of an individual's methylation data relative to the disorder-specific EpiSign™ probe sets and controls. MVP score assessment had a scale with thresholds of > 0.5 for positive, < 0.1 negative, 0.1–0.5 inconclusive or moderate confidence. A detailed description of this analytics protocol was described previously[25,87]. Possible types of results included: positive (matching an EpiSign™ disorder), negative (not matching any EpiSign™ disorder), and inconclusive (described in detail in results).

### Exome and genome sequencing

If sequencing data were already available for the individual on a collaborative research basis, these data were reviewed. If the data were unavailable, ES or GS was performed on peripheral blood-derived DNA using standard Illumina short-read sequencing techniques and bioinformatic approaches (Supplementary Methods). We validated potentially pathogenic variants with Sanger sequencing and confirmed sample identity and relatedness (e.g. trios) using Powerplex Short-Tandem Repeat (STR) Identification analysis.

### RNA-sequencing and gene expression analysis

RNA was extracted using the *Quick*-RNA Miniprep Kit (Zymo Research) from dermal fibroblasts established from skin punch biopsies for Family 2 ($n = 2$) and Family 3 ($n = 3$) described in the results. RNA-seq was performed using standard Illumina short-read sequencing practices (Supplementary Methods), and the reads were processed using the "nfcore/rnaseq" pipeline. Removal of the adapter sequences was performed using Trim Galore!, and low-quality reads were eliminated with FastQC[88]. Subsequently, reads were aligned to a reference genome using the STAR aligner[89]. Gene expression quantification was performed using Salmon[90], which estimates transcript abundance. To determine gene "dropout," the OUTRIDER algorithm[36] was applied to RNA-seq data for Family 2 (proband and mother), Family 3 (proband and father), and Family 3 (mother and father) against a publicly available dataset of $n = 139$ fibroblast samples[37]. PCA displayed no batch groupings, and genes with Fragments Per Kilobase of transcript per Million mapped reads (FPKM) < 1 were removed as lowly expressed

genes. Results were considered significant if they had a *padj* < 0.05 and a z-score cutoff of ± 2.

### Refinement of a CHD2 episignature

A total of 17 females and 12 males with genetic variants in *CHD2* and clinical features consistent with *CHD2*-epileptic encephalopathy of childhood (EEOC) were included in this expanded 850 K cohort. The detailed list of genetic variants classified as pathogenic or likely pathogenic according to the American College of Medical Genetics guidelines is in Supplementary Data 2B. All samples and records were deidentified.

Details of the methylation data analysis and episignature refinement are as previously described[25,52,60,91]. Briefly, methylation signal intensities were imported into R 4.1.3 for analysis. Normalization was performed by the Illumina normalization method with background correction using minfi[68]. Probes located on X and Y chromosomes, known SNPs, or probes that cross-react were excluded[67,69]. Samples containing failed probes of more than 5% ($p > 0.1$, calculated by the minfi package) were also removed. The genome-wide methylation density of all samples was examined, and principal component analysis (PCA) was performed to visualize the overall data structure of the batches and to identify outlier samples. All 29 samples passed and were used for probe selection. The MatchIt package was used to randomly select controls, which were matched for age, sex, and array type from the EKD at the LHSC, as previously described in refs. [25,92]. The methylation level of each probe was calculated as the ratio of methylated signal intensity over the sum of methylated and unmethylated signal intensities (β-values), ranging between 0 (completely unmethylated) and 1 (fully methylated). β-values were then converted to M-values by logit transformation using the formula $\log_2(\beta/(1-\beta))$ to perform linear regression modeling, which was used to identify the differentially methylated probes (DMPs), via the R package limma[93]. The analysis was also adjusted for blood cell-type compositions, using the Houseman algorithm[94]. The estimated blood cell proportions were added to the model matrix of the linear models as confounding variables. The generated *p*-values were moderated using the eBayes function in the limma package and were corrected for multiple testing using the Benjamini and Hochberg (BH) method.

Following this, probe selection was performed in three steps. Firstly, 1000 probes were selected, which had the highest product of methylation difference means between case and control samples and the negative of the logarithm of multiple-testing corrected *p* values derived from the linear modeling. Secondly, a receiver's operating characteristic (ROC) curve analysis was performed, and 200 probes with the highest area under the ROC curve (AUC) were retained. Lastly, probes having pair-wise Pearson's correlation coefficient greater than 0.85 within case and control samples separately were removed (none of the selected 200 probes met this criteria). This resulted in the identification of 200 DMPs. These probes were used for the construction of a hierarchical clustering model using Ward's method on Euclidean distance, as well as a MDS model by scaling of the pairwise Euclidean distances between samples.

### Functional annotation and correlation of the CHD2 episignature

Functional annotation and episignature cohort comparisons were performed according to our published methods[87]. Briefly, to assess the percentage of DMPs shared between the CHD2 episignature and other neurodevelopmental conditions on the EpiSign™ clinical classifier, heatmaps and circos plots were produced. Heatmaps were plotted using the R package pheatmap (version 1.0.12) and circos plots using the R package circlize (version 0.4.15)[95]. To determine the genomic location of the DMPs, probes were annotated in relation to CGIs and genes using the R package annotatr[49] with AnnotationHub and annotations hg19_cpgs, hg19_basicgenes, hg19_genes_intergenic, and hg19_genes_intronexonboundaries. CGI annotations included CGI

shores from 0–2 Kb on either side of CGIs, CGI shelves from 2–4 Kb on either side of CGIs, and inter-CGI regions encompassing all remaining regions. A chi-squared goodness of fit test was performed in R to investigate the significance between background DMP annotation distribution and the CHD2 cohort annotation distribution. *P* values were obtained for both annotation categories (gene and CGIs). To assess the relationship between the expanded 850 K only CHD2 cohort and other EpiSign™ disorders, the distance and similarities between cohorts were analyzed using clustering methods and visualized on a tree and leaf plot. This assessed the top 500 DMPs for each cohort, ranked by *p*-value. For cohorts with less than 500 DMPs, all DMPs were used. Tree and leaf plots, generated using the R package TreeAndLeaf[96], illustrated additional information, including global mean methylation difference and total number of DMPs identified for each cohort.

### Whole-genome bisulfite sequencing

Genomic peripheral blood-derived DNA from $n = 3$ CHD2 trios (proband and parents) and $n = 1$ CHD2 singleton (proband) (total $n = 10$ samples) were bisulfite-converted and then underwent WGBS using standard Illumina short-read sequencing processing methods (Supplementary Methods). Reads were trimmed by Trim Galore! and aligned to the GRCh38/hg38 human genome reference using BSMAP2.74. The methylation ratios from BSAMP mapping results were extracted using methratio.py. Duplicated reads were removed and CpG methylation from both strands was combined. The methylation ratios were also corrected according to the C/T SNP information estimated by the G/A counts on reverse strand.

### DMR calling of DNA methylation array and WGBS

We performed DMR analysis on Illumina 850 K EPIC methylation array data for 16 individuals with DEEs harboring pathogenic variants in *CHD2* compared to 18 controls. The data were normalized using the minfi package's functional normalization algorithm[97], and we employed two independent R packages to call DMRs, bumphunter[44] and DMRcate[45]. DMRs were defined as those passing a significance threshold of $p < 0.05$ for bumphunter and Fisher's multiple comparison $P < 0.05$ for DMRcate. A minimum of three CpGs and mean methylation difference between CHD2 and controls of at least 5% was also required (bumphunter "cutoff" and DMRcate "betacutoff"= 0.05) in either the hyper or hypo direction. For bumphunter, smoothing was used, and the number of permutations for each condition was set to B = 1000. For DMRcate, default settings were used, and the Gaussian kernel bandwidth for smoothed-function estimation was set to λ = 1000, meaning that significant CpGs further than 1000 nucleotides were in separate DMRs.

The methylCall data from WGBS, which consists of the total number of reads covered for each CpG site and the number of methylated C's at each CpG site, was used for calling DMRs between four individuals with DEEs caused by pathogenic *CHD2* variants and six unaffected parents. Firstly, CpG sites with less than 10X coverage and those on the sex chromosomes were removed. DMRs were called from WGBS methylCall data using two independent R packages, DMRcate[48] and DSS[47]. DMRcate identifies and ranks the most differentially methylated regions across the genome, while DSS detects differentially methylated loci or regions from WGBS. For DMRcate, the scaling factor for bandwidth "C" was set to 50, as recommended for WGBS. DSS was run with default parameters. DMRs were defined by each algorithm (with smoothing) as regions of a minimum of five CpGs with significance (Fisher's multiple comparison *P* value < 0.05) and minimum methylation differences of 5% in either the hyper or hypo direction (DSS "delta" and DMRcate "betacutoff"=0.05) between cases and controls.

The genomic locations of output DMR calls were intersected between both callers requiring a minimum overlap of 50 bp in the

same direction to reduce the false positive rate. This resulted in high-confidence lists of DMRs predicted by two independent callers each for array (bumphunter and DMRcate) and WGBS (DMRcate and DSS). The methylation difference between CHD2 and control was averaged between both callers for the final DMR list. DMRs were segmented by mean methylation difference between CHD2 and control (5%, 10%, 15%, and 20%) for visualization and annotation with CpG elements (islands, shores, shelves) and gene regions (1–5 Kb upstream TSS, promoters as <1 Kb upstream TSS, 5'UTRs, exons, introns, and 3'UTRs) using annotatr[49]. To get adequate CpG element counting (i.e. a DMR spanning both a shore and shelf would not get counted twice), CpG annotations were adjusted for DMR size by calculating representation across CpG elements as a fraction of the total DMR length. Details for in-depth annotation and enrichment calculation of CHD2 epsignature probes and DMRs for regulatory elements (bivalent regions, enhancers, promoters, TFBS, and DNase sites) may be found in Supplementary Methods.

### Reporting summary
Further information on research design is available in the Nature Portfolio Reporting Summary linked to this article.

## Data availability
Methylation array data for individuals with uDEEs and those with pathogenic variants in *CHD2* who have given consent for data sharing is available through the Gene Expression Omnibus (GSE269416). Additional data requests can be directed to H.C.M.

## Code availability
The methylation array analysis pipeline used in part of this study for epivariant detection can be accessed on GitHub: https://github.com/stjude-biohackathon/MethylMiner. Further bash and shell scripts created for this manuscript and used in the analysis may be found on the Mefford Laboratory GitHub: https://github.com/MeffordLab/2024_GenomeWideMethylationPaper. EpiSign™ is proprietary commercial software and is not publicly available.

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

## Acknowledgements

We thank all the individuals and their families for participating in this research. Major funding for this project was provided by a grant (#631106) from Citizens United for Research in Epilepsy (CURE). A subset of DNA methylation arrays was provided by the University of Washington Center for Rare Disease Research (UW-CRDR), formerly known as the Center for Mendelian Genomics (CMG), with support from NHGRI grants U01 HG011744, UM1 HG006493, U24 HG011746, and with enthusiastic support from the late Debbie Nickerson. We gratefully acknowledge support from the Australian Epilepsy Research Foundation grant, the Australian National Health and Medical Research Council (NHMRC) Centre for Research Excellence Grant (GNT2006841), NHMRC Synergy Grant (GNT2010562), the Health Research Council of New Zealand, Cure Kids New Zealand, and the Estate of Ernest Hyam Davis and the Tedd and Mollie Carr Endowment Trust. We acknowledge the Epi25 Consortium, which provided exome sequence data for review for a subset of individuals. C.W.L. has been funded through the American Epilepsy Society (AES) predoctoral fellowship (#919453) and the St. Jude Graduate School of Biomedical Sciences. We would also like to acknowledge the inaugural St. Jude Biohackathon 2022 for coordinating the event that led to a team comprised of C.W.L., P.K., M.N.D., and W.R., who assembled the MethylMiner pipeline described here. K.L.P. has been funded through the *GRIN2B* foundation and CURE. The project was also supported by NIHR Manchester Biomedical Research Centre (NIHR203308) and the MRC Epigenomics of Rare Diseases Node (MR/Y008170/1); we thank Siddharth Banka and David Gokhale for their support. Research reported in this manuscript by M.W.H., S.K., H.D., K.C.W., J.A.R., and H.T.C., was supported by the NIH Common Fund, through the Office of Strategic Coordination/Office of the NIH Director under Award Numbers U01HG007709 and U01HG007942. I.E.S. is also supported by an NHMRC Senior Investigator Fellowship (GNT1172897). D.E.M. is supported by NIH grant DP5OD033357. We acknowledge Pratibha Kottapalli and Sanchit Trivedi from the St. Jude Hartwell Center, who performed Illumina sequencing for this project. The content is solely the responsibility of the authors and does not necessarily represent the official views of the NIH. Schematics featured in this manuscript were created with BioRender.com and released under a Creative Commons Attribution-Non-Commercial-Noderivs 4.0 International License (CC-BY-NC-ND).

## Author contributions

H.C.M. and C.W.L. conceptualized the study. C.W.L., H.C.M., B.S., C.R., I.E.S., D.E.M., L.G.S., and S.F.B. designed the research methodology. C.W.L., C.R., S.S., H.E.P., M.P.Z., J.G., S.B.G., D.M.N., M.H., E.V.W., D.D., P.K., M.N.D., W.R., H.M., J.K., M.A.L., R.R., and S.K. performed formal analysis. C.W.L., E.P.A., M.P.Z., J.G., N.L., M.H., E.V.W., and S.R.O. executed experimental investigation. H.C.M., C.W.L., UW-CRDR, and G.N. acquired funding. H.C.M, I.E.S., L.G.S., S.F.B., S.J.R., A.L.S., E.S.B., T.J.A., Z.W., G.L.C., H.T.C., J.A.R., K.C.W., H.D., M.W.H., D.L., T.L.S., K.L.P., UDN, G.C., N.C., L.D., D.G., G.L., T.R., D.S., M.L.T., M.A., S.G., and E.A.J. recruited, phenotyped, and provided samples or data for individuals in this study. C.W.L., C.R., S.S., and H.E.P. prepared figures. C.W.L. and H.C.M. wrote the original draft. All authors reviewed and edited the manuscript. Clarification on abbreviations used to refer to the authors and full lists of consortia members (UW-CRDR and UDN) may be found in the Supplementary Information.

## Competing interests

B.S. is a shareholder in EpiSign Inc, a company involved in commercialization of EpiSign™ software. D.E.M. is on a scientific advisory board at ONT and has received travel support from ONT to speak on their behalf. D.E.M. is engaged in a research agreement with ONT. D.E.M. holds stock options in MyOme. I.E.S. has served on scientific advisory boards for BioMarin, Chiesi, Eisai, Encoded Therapeutics, GlaxoSmithKline, Knopp Biosciences, Nutricia, Rogcon, Takeda Pharmaceuticals, UCB, Xenon Pharmaceuticals, Cerecin; has received speaker honoraria from GlaxoSmithKline, UCB, BioMarin, Biocodex, Chiesi, Liva Nova, Nutricia, Zuellig Pharma, Stoke Therapeutics and Eisai; has received funding for travel from UCB, Biocodex, GlaxoSmithKline, Biomarin, Encoded Therapeutics Stoke Therapeutics and Eisai; has served as an investigator for Anavex Life Sciences, Cerevel Therapeutics, Eisai, Encoded Therapeutics, EpiMinder Inc, Epygenyx, ES-Therapeutics, GW Pharma, Marinus, Neurocrine BioSciences, Ovid Therapeutics, Takeda Pharmaceuticals, UCB, Ultragenyx, Xenon Pharmaceuticals, Zogenix and Zynerba; has consulted for Care Beyond Diagnosis, Epilepsy Consortium, Atheneum Partners, Ovid Therapeutics, UCB, Zynerba Pharmaceuticals, BioMarin, Encoded Therapeutics and Biohaven Pharmaceuticals; and is a Non-Executive Director of Bellberry Ltd and a Director of the Australian Academy of Health and Medical Sciences and the Australian Council of Learned Academies Limited. I.E.S. may accrue future revenue on pending patent WO61/010176 (filed: 2008): Therapeutic Compound; has a patent for *SCN1A* testing held by Bionomics Inc and licensed to various diagnostic companies; has a patent molecular diagnostic/theragnostic target for benign familial infantile epilepsy (BFIE) [PRRT2] 2011904493 & 2012900190 and PCT/AU2012/001321 (TECH ID:2012-009). L.G.S. receives funding from the Health Research Council of New Zealand and Cure Kids New Zealand, is a consultant for the Epilepsy Consortium, and has received travel grants from Seqirus and Nutricia. L.G.S. has received research grants and consultancy fees from Zynerba Pharmaceuticals and has served on Takeda and Eisai Pharmaceuticals scientific advisory panels. The Department of Molecular and Human Genetics at Baylor College of Medicine receives revenue from clinical genetic testing conducted at Baylor Genetics Laboratories. The remaining authors declare no competing interests.

## Additional information

Christy W. LaFlamme [1,2,43], Cassandra Rastin [3,4,43], Soham Sengupta[1], Helen E. Pennington [1,5], Sophie J. Russ-Hall [6], Amy L. Schneider [6], Emily S. Bonkowski[1], Edith P. Almanza Fuerte[1], Talia J. Allan [6], Miranda Perez-Galey Zalusky [7], Joy Goffena [7], Sophia B. Gibson [7,8], Denis M. Nyaga [9], Nico Lieffering[9], Malavika Hebbar[7], Emily V. Walker [10], Daniel Darnell[10], Scott R. Olsen[10], Pandurang Kolekar [11], Mohamed Nadhir Djekidel[12], Wojciech Rosikiewicz [12], Haley McConkey[4], Jennifer Kerkhof [4], Michael A. Levy[4], Raissa Relator[4], Dorit Lev[13], Tally Lerman-Sagie[14,15], Kristen L. Park [16], Marielle Alders [17], Gerarda Cappuccio[18,19], Nicolas Chatron [20,21], Leigh Demain[22], David Genevieve [23], Gaetan Lesca [20,21], Tony Roscioli [24,25,26], Damien Sanlaville[20,21], Matthew L. Tedder[27], Sachin Gupta[28], Elizabeth A. Jones[22,29], Monika Weisz-Hubshman[30,31], Shamika Ketkar[30], Hongzheng Dai[30], Kim C. Worley [30], Jill A. Rosenfeld [30], Hsiao-Tuan Chao [30,32,33,34,35,36], Undiagnosed Diseases Network*, Geoffrey Neale [10], Gemma L. Carvill [37], University of Washington Center for Rare Disease Research*, Zhaoming Wang [11,38], Samuel F. Berkovic [6], Lynette G. Sadleir[9], Danny E. Miller [7,39,40], Ingrid E. Scheffer [6,41,42], Bekim Sadikovic [3,4,44] ✉ & Heather C. Mefford [1,44] ✉

[1]Center for Pediatric Neurological Disease Research, Department of Cell and Molecular Biology, St. Jude Children's Research Hospital, Memphis, TN 38105, USA. [2]Graduate School of Biomedical Sciences, St. Jude Children's Research Hospital, Memphis, TN 38105, USA. [3]Department of Pathology & Laboratory Medicine, Western University, London, ON N5A 3K7, Canada. [4]Verspeeten Clinical Genome Centre, London Health Science Centre, London, ON N6A 5W9, Canada. [5]Department of Mathematics & Statistics, Rhodes College, Memphis, TN 38112, USA. [6]Epilepsy Research Centre, Department of Medicine, University of Melbourne, Austin Health, Heidelberg, VIC 3084, Australia. [7]Division of Genetic Medicine, Department of Pediatrics, University of Washington and Seattle Children's Hospital, Seattle, WA 98195, USA. [8]Department of Genome Sciences, University of Washington School of Medicine, Seattle, WA 98195, USA. [9]Department of Paediatrics and Child Health, University of Otago, Wellington 6242, New Zealand. [10]Hartwell Center for Bioinformatics and Biotechnology, St. Jude Children's Research Hospital Memphis, Memphis, TN 38105, USA. [11]Department of Computational Biology, St. Jude Children's Research Hospital,

Memphis, TN 38105, USA. [12]Center for Applied Bioinformatics, St. Jude Children's Research Hospital, Memphis, TN 38105, USA. [13]Institute of Medical Genetics, Wolfson Medical Center, Holon 58100, Israel. [14]Fetal Neurology Clinic, Pediatric Neurology Unit, Wolfson Medical Center, Holon 58100, Israel. [15]Sackler School of Medicine, Tel-Aviv University, Tel-Aviv, Israel. [16]Departments of Pediatrics and Neurology, University of Colorado School of Medicine, Aurora, CO 80045, USA. [17]Department of Human Genetics, Amsterdam Reproduction and Development Research Institute, Amsterdam UMC, University of Amsterdam, Amsterdam, Meibergdreef 9, Amsterdam, Netherlands. [18]Telethon Institute of Genetics and Medicine, Pozzuoli, Italy. [19]Department of Translational Medicine, Federico II University of Naples, Naples, Italy. [20]Department of Medical Genetics, Member of the ERN EpiCARE, University Hospital of Lyon and Claude Bernard Lyon I University, Lyon, France. [21]Pathophysiology and Genetics of Neuron and Muscle (PNMG), UCBL, CNRS UMR5261 - INSERM, U1315 Lyon, France. [22]Manchester Centre for Genomic Medicine, St Mary's Hospital, Manchester University NHS Foundation Trust, Health Innovation Manchester, Manchester, UK. [23]Montpellier University, Inserm Unit 1183, Reference Center for Rare Diseases Developmental Anomaly and Malformative Syndrome, Clinical Genetic Department, CHU Montpellier, Montpellier, France. [24]Neuroscience Research Australia (NeuRA), Sydney, NSW, Australia. [25]Prince of Wales Clinical School, Faculty of Medicine, University of New South Wales, Sydney, NSW, Australia. [26]New South Wales Health Pathology Randwick Genomics, Prince of Wales Hospital, Sydney, NSW, Australia. [27]Greenwood Genetic Center, Greenwood, SC 29646, USA. [28]TY Nelson Department of Neurology and Neurosurgery, The Children's Hospital at Westmead, Westmead, NSW, Australia. [29]Division of Evolution, Infection and Genomics, School of Biological Sciences, Faculty of Biology, Medicine and Health, University of Manchester, Manchester, UK. [30]Department of Molecular and Human Genetics, Baylor College of Medicine, Houston, TX 77030, USA. [31]Texas Children's Hospital, Genetic Department, Houston, TX 77030, USA. [32]Department of Pediatrics, Section of Neurology and Developmental Neuroscience, Baylor College of Medicine, Houston, TX 77030, USA. [33]Cain Pediatric Neurology Research Foundation Laboratories, Jan and Dan Duncan Neurological Research Institute, Texas Children's Hospital, Houston, TX 77030, USA. [34]Texas Children's Hospital, Houston, TX 77030, USA. [35]Department of Neuroscience, Baylor College of Medicine, Houston, TX 77030, USA. [36]McNair Medical Institute, The Robert and Janice McNair Foundation, Houston, TX 77030, USA. [37]Ken and Ruth Davee Department of Neurology, Northwestern University Feinberg School of Medicine, Chicago, IL, USA. [38]Department of Epidemiology and Cancer Control, St. Jude Children's Research Hospital, Memphis, TN 38105, USA. [39]Department of Laboratory Medicine and Pathology, University of Washington, Seattle, WA 98195, USA. [40]Brotman Baty Institute for Precision Medicine, University of Washington, Seattle, WA 98195, USA. [41]Department of Paediatrics, University of Melbourne, Royal Children's Hospital, Melbourne, VIC, Australia. [42]Florey Institute and Murdoch Children's Research Institute, Melbourne, VIC, Australia. [43]These authors contributed equally: Christy W. LaFlamme, Cassandra Rastin. [44]These authors jointly supervised this work: Bekim Sadikovic, Heather C. Mefford. *Lists of authors and their affiliations appear at the end of the paper.
✉ e-mail: Bekim.Sadikovic@lhsc.on.ca; Heather.Mefford@stjude.org

## Undiagnosed Diseases Network

Monika Weisz-Hubshman[30,31], Shamika Ketkar[30], Hongzheng Dai[30], Kim C. Worley[30], Jill A. Rosenfeld[30], Hsiao-Tuan Chao[30,32,33,34,35,36] & Danny E. Miller[7,39,40]

## University of Washington Center for Rare Disease Research

Miranda Perez-Galey Zalusky[7], Joy Goffena[7], Sophia B. Gibson[7,8], Danny E. Miller[7,39,40] & Heather C. Mefford[1,44]✉

A full list of members and their affiliations appears in the Supplementary Information.

