## [Peer Review File · Nature Communications]

Diagnostic Utility of DNA Methylation Analysis in Genetically Unsolved Pediatric Epilepsies and CHD2 Episignature RefinementREVIEWER COMMENTS

Reviewer #1 (Remarks to the Author):

The manuscript by LaFlamme et al. aimed to test the diagnostic utility of DNA methylation profiles by analyzing rare outlier DMRs and episignatures obtained from peripheral blood samples using 850K arrays in 516 genetically unsolved DEE cases. The study shows that rare outlier DMR analysis and ONT sequencing can pinpoint previously missed genetic variants in unsolved DEEs including balanced translocations, CG-rich repeat expansions, and CNVs, however with limited sensitivity and accuracy. The explainability of disease-related biological changes from the here described episignatures is very limited.

The following specific comments apply:

- The authors state that rare epivariants on single genes can drive disease through changes in gene expression whereas episignatures can serve as biomarkers for monogenic disorders in rare diseases. Please clarify the contradiction. Do the authors mean to say that „episignatures“ do not affect gene expression? Or that „epivariants“ cannot serve as biomarkers? This would not be true. For e.g. MGMT promoter methylation is a clinically widely used diagnostic and prognostic biomarker in glioma (Hegi et al, NEJM 2008). Also, check ll.148-150. Do genome-wide approaches fail to detect rare DMRs? Statements like „Rare epigenetic variations („epivariants“) disrupt normal methylation and cause disease,“ are misleading as they suggest there is no exception to that. Taken together, the writing lacks clarity and the terminology used seems not well defined also in the broader literature. Consider rephrasing to avoid ambiguity.
- Authors state that episignatures in diagnostics were first clinically validated and implemented in 2019 with the EpiSign Assay. This does not hold true as in cancer episignatures (if defined as an epigenetic signature specifically associated with a disease or pathology) have been used earlier than that. For example, the brain tumor classifier as a diagnostic tool was implemented and validated in 2018 (Capper et al., Nature 2018) and since then have been included in WHO diagnostic recommendations. If episignature is defined otherwise, please clarify.
- ll.164-165: It is stated episignature classifiers are blood-specific. However, there are clinically relevant classifiers using epigenetic signatures from other tissues (e.g. brain or other solid tissues). Please clarify whether episignature classifier is a unique term used only in neurodevelopmental disorders.
- Methods: Please provide more detail on sample and data collection. E.g., the authors state that methylation array data for healthy controls were drawn from a public database (l.183). They reference a publication (Ref 23) describing the PPMI study. Neither the cited manuscript nor the PPMI website provides information on DNA methylation data, sample distribution of the cohort, and data accessibility. No GEO accession number or similar data identifiers are available. Specify how many samples have been analyzed using which method/array type.
- During methylation QC analysis for 850K array data, only probes that failed detection in >10% of samples and probes overlapping with common SNPs were removed (ll.218-219). However, probes not uniquely matching and cross-reactive probes were not excluded. Please justify. Also explain, why the methodology later on in the CHD2 episignature refinement approach differs for 850K data analysis (ll.349-351).
- Episignature testing was performed using the EpiSign platform, a test that can detect multiple methylation abnormalities in 64 genes (according to EpiSign website) associated with certain imprinting or triplet repeat conditions. However, the authors state the assay identifies 90 episignatures representing 70 disorders associated with 96 genes. Please briefly clarify for the reader the difference in gene numbers and the sensitivity and specificity of MVP scores for their association with an EpiSign disorder. There were two individuals in the study with clear genetic and clinical findings that were inconclusive for their episignature. The authors however state that the inconclusive episignatures (i.e., below cutoff, <0.5) matched the disorder. Likewise, one individual with high-confidence episignature findings had a genetic variant in a different gene than predicted. Another 40 unsolved individuals and 9 controls had inconclusive results of episignatures. So the question is, what is the positive and

negative predictive value of these signatures? Also, any suggested diagnostic or predictive value of episignatures in unsolved DEEs should be validated in an independent cohort to confirm the robustness and reproducibility of the identified methylation patterns.

- If the top 27 most implicated genetic causes of DEEs explain 80% of DEEs, but only 1/27 (i.e., CHD2) has a so-called „episignature“, the diagnostic value of these signatures, in the limited sense as discussed in the manuscript, appears neglectable in this patient group.

- The authors use the episignature and DMRs of CHD2 DEE patients compared to controls to infer the biology of the disease. The signature was obtained from blood, however, the main presenting symptom, i.e., seizures, is a malfunction of the brain. While there is a claim that disease-related episignatures are stable across different tissues, the general overlap of DNA methylation profiles between blood and brain is as low as 5%. Moreover, the episignatures comprise 200 individual DMPs, i.e., individual CpG sites, which imposes several questions regarding their significance for the regulation of gene expression. This could be partially addressed if overlap with TF binding sites or other regulatory features could be demonstrated. Alternatively, the authors use an array and additionally, a WGBS DMR approach identifying more DMRs than DMPs, however only partial overlaps and limited links to gene functions. Results are only displayed in the supplements and not discussed.

- Are there any implications of episignatures for treatment?

Reviewer #2 (Remarks to the Author):

General comments

Laflamme, Costain and their co-investigators have performed a methylation analysis of DNA from a cohort of 516 individuals with developmental epileptic encephalopathy. The individuals in the study had clinical testing already performed and they were “unsolved” at the time of the methylation analysis. Methylation analysis was able to identify unique methylation signatures for 10 individuals (about 2%).

The study is novel and highlights the utility of methylation analysis as part of the diagnostic pathway in rare disease. The sample is large (516 unsolved) for a study of rare disease and highlights the work by recognized experts in the field (both in epilepsy genetics and in methylation studies). The results are compelling.

The researchers show the ability of methylation studies to identify patients with differences in methylation at single loci. As a result, the researchers show evidence for repeat expansion in three genes causing differences in methylation (long read sequencing was performed after methylation differences were observed). These genes are BCLAF3, DIP2B and CSNK1E. The expansions have not been reported with DEE before (I do see that there has been 1 paper on DIP2B published over 15 years ago).

Further work will be needed to show that these are bone fide disease genes and will likely require methylation studies as a first step to identify patients who have an underlying expansion – followed by expansion detection. The investigation into the DMRs (differentially methylated regions) also highlighted the ability for the analysis to identify copy number variants and complex rearrangements.

Of particular interest, was a screen of the 516 unsolved with the known 70 validated signatures (via EpiSign). The known signatures identified genetic diagnoses in 1% in the undiagnosed cohort. When combined with the assessment of rare, outlier DMRs the solve rate was about 2%.

While the overall solve rate was low, this was a sample that had previously undergone extensive testing. If the methylation assay had been performed earlier in the diagnostic journey, the results would have likely shown a much higher “solve rate”.

The researchers also perform further studies in 29 individuals with CHD2 variation and further refine

an already-known methylation signature in DEE94.

The researchers appropriately comment on a major limitation, that the most common causes of DEE are under-represented with known, validated epesignatures. They state that the only validated signature in DEE is CHD2 (and hence they studied this further).

This is an important limitation though, in time, other signatures will be validated for the DEE's.

Discussion is well written and balanced and discusses the benefits and challenges for this emerging technology in rare disease.

Supplemental is well written and I have no concerns.

Major comments

I have none. This is an important manuscript that will be important for all individuals studying the underlying causes of epilepsy and encephalopathies. Methylation will be an important part of the diagnostic process for the unsolved.

Minor comments only

Introduction: Can the authors provide a reference for the sentence, "This, in turn, improves outcomes but is not possible when the etiology is unknown ("unsolved")."

Introduction: Can the authors add reference for "Episignatures have been found for neurodevelopmental disorders where epilepsy is part of the phenotype...."

Material and methods: Can you provide numbers with the percent, 80% had a gene panel, 40% microarray analysis, 76% ES, and 38% GS. Collectively, 98% had at least 193 one sequence-based investigation (gene panel, ES, or GS)

Results: Can they include the number of repeats detected by LRS for the variants in CSNK1E, DIP2B and BCLAF3

Line 641: "An additional 40 individuals with DEEs (80% unsolved) and....". I suspect the 80% is a typo and should be 8% (40/516).

Response to Reviewer's Comments

We thank the editor and the reviewers for their time and feedback on our study. We have addressed the comments and believe the updates have greatly improved the overall quality of the manuscript.

Since our initial submission, we have analyzed data for an additional 66 individuals with unsolved DEEs, thereby increasing the cohort from 516 to 582 unsolved DEE cases. In this revised manuscript, we report two novel findings from this additional analysis. The first is rare hypermethylation of the *STX1B* TSS and promoter of an individual with unsolved DEE (new Figure 3). Through genome sequencing, we identified an underlying likely pathogenic copy number deletion encompassing part of the promoter, TSS, exon 1, and part of intron 1 of *STX1B* that is consistent with phenotype and segregates with disease in multiple affected family members.

The second is an episignature finding of SMS in a proband with unsolved DEE. Variants in SMS cause an X-linked intellectual developmental disorder. Exome sequencing revealed a maternally inherited pathogenic missense variant (see revised manuscript, Supplementary Figure 23). This did not change the overall yield of 2% for methylation analysis and, in fact, validated the originally reported diagnostic yield on an independent cohort (n=66).

Figure 3 (new): Rare outlier DMR analysis identifies copy number deletion in a family with GEFs+.

The statements and comments from reviewers #1 and #2 are listed below in **bold**. Our responses are located below each comment. Portions taken from the manuscript are indented and *italicized*, where major changes to the text are noted in **red**. Corrected typos or minor changes to the text that are not associated with any major updates or addressing any reviewer's comments (e.g. changing "Figure S1" to "Supplementary Figure 1") are not highlighted for simplicity of the revised manuscript.

Reviewer #1 (Remarks to the Author):

The manuscript by LaFlamme et al. aimed to test the diagnostic utility of DNA methylation profiles by analyzing rare outlier DMRs and episignatures obtained from peripheral blood samples using 850K arrays in 516 genetically unsolved DEE cases. The study shows that rare outlier DMR analysis and ONT sequencing can pinpoint previously missed genetic variants in unsolved DEEs including balanced translocations, CG-rich repeat expansions, and CNVs, however with limited sensitivity and accuracy. The explainability of disease-related biological changes from the here described episignatures is very limited.

The following specific comments apply:

The authors state that rare epivariants on single genes can drive disease through changes in gene expression whereas episignatures can serve as biomarkers for monogenic disorders in rare diseases. Please clarify the contradiction. Do the authors mean to say that „episignatures“ do not affect gene expression? Or that „epivariants“ cannot serve as biomarkers? This would not be true. For e.g. MGMT promoter methylation is a clinically widely used diagnostic and prognostic biomarker in glioma (Hegi et al, NEJM 2008).

The reviewer raises a very good point that the language of this statement seems contradictory. It is true that epivariants can serve as diagnostic and prognostic biomarkers, and episignatures may affect gene expression (relatively unstudied). We have updated this phrasing in the abstract to present a more refined definition of epivariants and episignatures.

[Lines 103-104]: *“Rare epigenetic variations (“epivariants”) can drive disease by modulating gene expression at single loci. DNA methylation changes **at many loci across the genome** can result in distinct “episignatures” biomarkers for monogenic disorders in a growing number of rare diseases.”*

Also, check II.148-150. Do genome-wide approaches fail to detect rare DMRs? Statements like „Rare epigenetic variations („epivariants“) disrupt normal methylation and cause disease,“ are misleading as they suggest there is no exception to that. Taken together, the writing lacks clarity and the terminology used seems not well defined also in the broader literature. Consider rephrasing to avoid ambiguity.

We thank the reviewer for the feedback and have updated the wording to avoid ambiguity.

[Lines 137-138]: *“Rare epigenetic variations (“epivariants”) are part of human genetic variation, but in some cases have been shown to disrupt normal methylation and cause disease.”*

Authors state that episignatures in diagnostics were first clinically validated and implemented in 2019 with the EpiSign Assay. This does not hold true as in cancer episignatures (if defined as an epigenetic signature specifically associated with a disease or pathology) have been used earlier than that. For example, the brain tumor classifier as a diagnostic tool was implemented and validated in 2018 (Capper et al., Nature 2018) and since then have been included in WHO diagnostic recommendations. If episignature is defined otherwise, please clarify.

We meant to be referring to implementation for rare monogenic disorders, but the reviewer is correct that epigenetic profiles were first used in cancer diagnostics, which should be acknowledged. We now acknowledge this in the introduction with the appropriate reference.

[Lines 153-154]: *“These epigenetic profiles were first implemented for cancer diagnostics with the introduction of the brain tumor classifier in 2018.”*

II.164-165: It is stated episignature classifiers are blood-specific. However, there are clinically relevant classifiers using epigenetic signatures from other tissues (e.g. brain or other solid tissues). Please clarify whether episignature classifier is a unique term used only in neurodevelopmental disorders.

It is important to recognize the DNA methylation-based classifiers are used for tissues other than blood (Lines 153-154 above). We have clarified the definition of episignatures for diagnosing rare diseases from blood.

[Lines 154-159]: *“A growing number of rare diseases exhibit these methylation patterns, or “episignatures,” in the blood that are reproducible among individuals with pathogenic variants within the same protein domain, gene, or protein complex, yielding highly sensitive and specific biomarkers (21,22). Since episignatures in diagnostics of rare neurodevelopmental disorders were first clinically validated and implemented with the EpiSign™ assay in 2019 (23), episignatures for nearly 70 rare diseases have been published. Episignatures provide strong evidence for genetic diagnosis, regardless of whether an underlying pathogenic DNA variant is identified, and to resolve variants of uncertain significance (VUS).”*

[Lines 169-170]: *“Episignature classifiers for rare diseases are trained on data obtained from blood-derived DNA and are, therefore, blood-specific.”*

Methods: Please provide more detail on sample and data collection. E.g., the authors state that methylation array data for healthy controls were drawn from a public database (I.183). They reference a publication (Ref 23) describing the PPMI study. Neither the cited manuscript nor the PPMI website provides information on DNA methylation data, sample distribution of the cohort, and data accessibility. No GEO accession number or similar data identifiers are available. Specify how many samples have been analyzed using which method/array type.

The PPMI data are publicly available but require an application to access the datasets and associated information about the protocols. A brief paragraph about acquiring this dataset may now be found in the Supplementary Materials and Methods.

[Lines 270-275, Supplementary]: *“A short application for access to the PPMI dataset is required. This may be found by navigating to <https://www.ppmi-info.org/> and choosing to “apply for data access.” Once granted, details about the sample collection and methylation array data acquisition are given as “genetic data” under “methylation profiling” in a PDF called <Project 140: Comprehensive Methylation Profiling Methods.pdf>. Sample distribution and other qualitative metadata for the cohort may be found in “study data” under “subject characteristics.”*

The number of PPMI samples analyzed (111 individuals before QC, and 110 individuals after QC) and array type are included in the text and Supplementary Table 1.

[Lines 253-256, Supplementary]: *“Unaffected, presumably healthy controls without DEEs include 111 healthy controls obtained through the Parkinson's Progression Markers Initiative (PPMI)(13), institutionally available data for 335 community control individuals without cancer from the St. Jude Life (SJLIFE) study(14), and 29 unaffected parents or siblings of participants in our study cohort (Supplementary Table 1).”*

The number of samples and method/array type are listed in Table S1. All data was collected using the EPIC 850K v1.0 platform. Previously, “EPIC 850K” was listed in Table S1. We have updated the list to “EPIC 850K v1.0” to further clarify further the exact version of the methylation array used for this study.

During methylation QC analysis for 850K array data, only probes that failed detection in

>10% of samples and probes overlapping with common SNPs were removed (II.218-219). However, probes not uniquely matching and cross-reactive probes were not excluded. Please justify. Also explain, why the methodology later on in the CHD2 episignature refinement approach differs for 850K data analysis (II.349-351).

Our methylation quality control workflow for rare DMR analyses was designed following previous protocols that perform similar outlier-based analyses (PMID: 29802345, PMID: 32937144). The requirement of 3 probes in a row for having outlier methylation values mitigates much of the possibility of interference from individual non-unique probes or cross-reactive probes (PMID:23314698 and PMID:27717381). To ensure that the analysis presented is robust and to address reviewer #1's comments, we have re-analyzed the data after excluding cross-reactive probes, and now acknowledge this change to the methods:

[Lines 226-228]: *“Individual CpG probes that failed (detection $p > 0.01$) in >10% of samples; also, probes overlapping with common SNPs and those previously reported as cross-reactive were removed (32,34).”*

We had previously excluded non-unique and cross-reactive probes in our episignature analyses of the unsolved DEE cohort, and we now emphasize this in methods:

Text in reference to the episignature testing of the unsolved cohort:

[Lines 310-314]: *“The data analysis pipeline was adapted from previously described methods (22) as summarized in Supplementary Figure 1A. Importantly, probes with a detection p -value > 0.01 , probes located on the X and Y chromosomes, probes that contained SNPs at the CpG interrogation or single-nucleotide extension sites, and probes that are known to cross-react with other genomic locations were removed (32,34).”*

Cross-reactive probes were previously addressed for the CHD2 episignature refinement. We have now added further citations:

[Lines 365-366]: *Probes located on X and Y chromosomes, known SNPs, or probes that cross-react (as reported by Illumina) were excluded (32,34).*

To address the second portion of reviewer #1's comment, the methodology for the epivariant (DMR) portion of the paper was performed at St. Jude Children's Research Hospital (correspondence: Heather Mefford). The CHD2 episignature refinement portion of the manuscript was performed at Western University (correspondence: Bekim Sadikovic). Since the methodologies slightly differ between the institutions, we reported them separately and have included both for full transparency.

Episignature testing was performed using the EpiSign platform, a test that can detect multiple methylation abnormalities in 64 genes (according to EpiSign website) associated with certain imprinting or triplet repeat conditions. However, the authors state the assay identifies 90 episignatures representing 70 disorders associated with 96 genes. Please briefly clarify for the reader the difference in gene numbers and the sensitivity and specificity of MVP scores for their association with an EpiSign disorder. There were two individuals in the study with clear genetic and clinical findings that were inconclusive for their episignature. The authors however state that the inconclusive episignatures (i.e., below cutoff, <0.5) matched the disorder. Likewise, one individual with high-confidence episignature findings had a genetic variant in a different gene than predicted. Another 40 unsolved individuals and 9 controls had inconclusive results of episignatures. So the question is, what is the positive and negative predictive value of these signatures? Also, any suggested diagnostic or predictive value of episignatures in unsolved DEEs should be validated in an independent cohort to confirm the robustness and reproducibility of the identified methylation patterns.

We thank the reviewer for the thorough review and provide further details explaining various parameters and related performance of the EpiSign™ classifier. For this study, we used version 4 (v4) of the EpiSign™ classifier (https://episign.lhsc.on.ca/img/EpiSign_v4_Menu.pdf); details pertaining to all of the assays, including genes/gene regions/CNVs, that are currently validated for use in clinical laboratories are in the linked pdf. We have previously described details summarizing the technical validation of EpiSign software (PMID: 35047860; PMID: 32109418) along with dozens of other manuscripts describing episignature validations for individual episignature cohorts. Clinical validation data and the initial diagnostic performance of the EpiSign classifier were described in detail (PMID: 33637969). Indeed, in addition to the technical validations of episignature algorithms, EpiSign algorithm is also clinically validated using blinded positive and negative reference cohorts as required for implementation in clinical testing environments. It is important to highlight that both reference data sets (based on patient and control cohorts within the EpiSign Knowledge Database [EKD]) and reference algorithms, and reference probe sets evolve over time as the EKD expands, which requires iterative validations and versioning of the clinical EpiSign classifier. The clinical validation and technical performance of v1-v3 classifiers was previously described (PMID: 33637969); the current v4 classifier performance using similar validation procedure with over 100 samples yielded ≥95% measures for sensitivity, specificity, positive predictive value and negative predictive value (unpublished data).

Regarding an individual “with high-confidence episignature findings *who* had a genetic variant in a different gene than predicted,” we recognize that the episignature gene may not be the primary driver of this individual’s disease. However, we want to recognize that it is unknown whether the variant could be contributing:

[Lines 688-689]: “*Thus, while it is unlikely that this KDM2B variant explains the individual’s phenotype, it still represents an underlying DNA change detected through episignature screening, and it remains possible that it has a modifying effect on phenotype.*”

Findings in this study, including the ~5% inconclusive result rate, are consistent and within the range of our previous reports (PMID: 33637969; 1.9% based on the first 200 patients tested), as well as the more current data (PMID: 38251460) describing 2,400 cases tested through the clinical EpiSign network showing the inconclusive result rate for complete analysis (8.6%, 144/1667), then the targeted analysis (2.3%, 17/732). Inconclusive findings are caused by methylation profiles that partially overlap existing episignatures that may be associated with hypomorphic, mosaic, or partially overlapping functional impact variant in the related gene(s). Alternatively, inconclusive episignature results in absence of related clinical features may indicate a partially overlapping episignature in a gene that is not yet clinically defined, or has not yet been included in EKD reference cohorts, or other factors such as environmental exposures with potential epigenetic effects.

[Lines 691-693]: *An additional 40 individuals with DEEs (n=32 unsolved, n=8 solved) and nine controls had inconclusive results for episignatures, consistent with the rate of inconclusive results in previous studies (79).*

Given that we are not testing new or previously undescribed episignatures and that we are using a clinically validated platform, we are confident in the sensitivity, specificity, PPV and NPV for this study. Although we continue to identify and test unsolved DEE cases through our research collaborations, to truly replicate this study in a similarly sized cohort would likely require an additional 2-3 years and >\$100,000. As described in the introduction, we were able to add data for an additional 66 unsolved DEE cases, with the same rate of discovery for both DMRs and episignature findings.

If the top 27 most implicated genetic causes of DEEs explain 80% of DEEs, but only 1/27 (i.e., CHD2) has a so-called „episignature“, the diagnostic value of these signatures, in the limited sense as discussed in the manuscript, appears neglectable in this patient group.

We agree that the diagnostic yield of episignatures for DEEs may increase as more episignatures are identified for DEE genes. DEEs are both phenotypically and genetically heterogeneous and can be caused by variations in hundreds of different epilepsy genes (PMID: 23934111). DEE-like and overlapping phenotypes exist for many genes for which episignatures have been derived. These genes include *CHD2* (direct DEE gene in the top 27 most common), *KDM5C*, *KDM2B*, *SETD1B*, *KMT2A*, *SMARCA2*, *ANKRD11*, *TET3*, and *UBE2A* for instance. Therefore, we set out to determine the diagnostic yield of episignatures for unsolved DEE with the caveat that some of the most common DEE genes do not yet have known episignatures. Although we are working to determine whether other DEE genes have episignatures, this requires collecting samples from enough patients with each rare disease to both (i) test and (ii) clinically validate each signature, work that is ongoing but beyond the scope of this study.

The authors use the episignature and DMRs of CHD2 DEE patients compared to controls to infer the biology of the disease. The signature was obtained from blood, however, the main presenting symptom, i.e., seizures, is a malfunction of the brain. While there is a

claim that disease-related episignatures are stable across different tissues, the general overlap of DNA methylation profiles between blood and brain is as low as 5%.

We do not claim “that disease-related episignatures are stable across different tissues.” We do make that statement for DMRs, as this has also been shown in previous studies:

[Lines 168-169]: *“Rare DMRs derived from individuals with ND-CA are recapitulated across multiple tissue types, including blood and fibroblasts (19).*

Because rare DMRs are often due to underlying constitutive DNA variations (such as CG-rich repeat expansions, as we show), they are often present throughout all cells in the body. However, we do not state that episignatures (collections of methylation for individual CpG sites across the genome) are stable in different tissue types. On the contrary, we state:

[Lines 169-170]: *“Episignature classifiers **for rare diseases** are trained on data obtained from blood-derived DNA and are, therefore, blood-specific.”*

As such, we agree with reviewer #1 that blood episignatures, which are associated with individual CpG sites, are unlikely to be fully recapitulated in the brain due to the strong tissue-specific effects. Given the lack of surgically resected or post-mortem brain tissue for patients with CHD2-related disorders, this is difficult to test. We, therefore, sought to interrogate DMRs between CHD2 and controls from blood DNA to provide deeper biological insights into the CHD2 blood episignature given that DMRs are more likely to affect gene expression and potential function than individual CpG sites. We recognize that any findings in the blood will still be limited and now highlight this limitation better in the results:

[Lines 790-793]: *“**Although CHD2 episignature and DMR insights are limited to the blood in our study, this work supports further investigations into CHD2 methylation of brain-relevant tissue types, such as cultured neurons, brain organoids or, when available, post-mortem tissue.**”*

Moreover, the episignatures comprise 200 individual DMPs, i.e., individual CpG sites, which imposes several questions regarding their significance for the regulation of gene expression. This could be partially addressed if overlap with TF binding sites or other regulatory features could be demonstrated. Alternatively, the authors use an array and additionally, a WGBS DMR approach identifying more DMRs than DMPs, however only partial overlaps and limited links to gene functions. Results are only displayed in the supplements and not discussed.

Based on the reviewer's suggestion, we have now annotated Table S8 (which is a list of all epismature probes and CHD2 DMRs called from array and WGBS) with nearest gene-based annotations using the HOMER annotation tool. Additionally, we have addressed reviewer 1's comment to annotate Table S8 for functional elements. To do this, we used the GREEN-DB (PMID: 35234913) collection of transcription factor binding site (TFBS from UCSC genome browser tracks), DNase peaks (from UCSC genome browser tracks), and multiple regulatory elements (bivalent regions, enhancers, and promoters which are drawn from various sources including ENCODE, FANTOM5, DiseaseEnhancers, BENGI, DECRES, etc.). Links to these database files are now provided in the Supplementary Materials and Methods.

We found that the CHD2 epismature probes and DMRs (n=4767 regions) are enriched for bivalent regions, enhancers, promoters, TFBS, and DNase sites, when compared to three independently generated randomized sets of regions (n=4767 regions) of varying, comparable lengths (50-3100bp) across the genome (simulating background). Enrichment was calculated as a ratio of the proportion of features in real regions divided by the total number of real regions, normalized by the proportion of features in simulated regions divided by the total number of simulated regions. Fischer's Exact *P* values were calculated using R for each condition and determined to be $P < 2.2e^{-16}$ for all conditions.

Figure 6: The CHD2 Epismature is associated with DMRs enriched in regulatory regions.

To address review 1's comment about the related figures being displayed only in the Supplementary and not discussed, we have now added **Figure 6: The CHD2 Epismature is associated with DMRs enriched in regulatory regions** to the main portion of this manuscript for more in-depth discussion. Figure 5 includes what was previously Supplementary Figure 31, where we zoom in to an example of multiple CHD2 epismature probes overlapping with WGBS DMRs. The number of regions annotated as functional compared to the simulated background regions are plotted in Figure 5B (bivalent regions, enhancers, promoters), 5C (TFBS), and 5D (Dnase sites).

Importantly, before this study, little work had been done to interrogate episignatures using WGBS data. Our work provides a landscape for further study of the CHD2 episignature, DMRs, and functional effects in disease-relevant tissue types, such as neuronal models. Indeed, we are pursuing such studies in patient-derived neurons (from iPSC), but this work is in early stages and beyond the scope of this manuscript.

Are there any Implications of episignatures for treatment?

Currently, there are no specific treatments for neurodevelopmental disorders based on episignatures. Insights from investigating how episignatures relate to underlying function may reveal potential therapeutic vulnerabilities. This has been a motivation for studying the CHD2 episignature in great depth to better understand how individual CpG probes might illuminate effects on gene regulation through larger DMRs. There are FDA-approved drugs, primarily for cancer therapies, that target epigenetic regulators (PMID: 20944599). With a better understanding of episignatures, there is potential to leverage insights toward applying therapies that directly target DNA methylation to treat these disorders.

We appreciate all the constructive comments from reviewer #1, which we believe have led to improvements in the manuscript. We hope our updates to this revised manuscript and the detailed rationale provided here have addressed the reviewer's concerns.

Reviewer #2 (Remarks to the Author):

General comments

Laflamme, Costain and their co-investigators have performed a methylation analysis of DNA from a cohort of 516 individuals with developmental epileptic encephalopathy. The individuals in the study had clinical testing already performed and they were "unsolved" at the time of the methylation analysis. Methylation analysis was able to identify unique methylation signatures for 10 individuals (about 2%).

The study is novel and highlights the utility of methylation analysis as part of the diagnostic pathway in rare disease. The sample is large (516 unsolved) for a study of rare disease and highlights the work by recognized experts in the field (both in epilepsy genetics and in methylation studies). The results are compelling.

The researchers show the ability of methylation studies to identify patients with differences in methylation at single loci. As a result, the researchers show evidence for repeat expansion in three genes causing differences in methylation (long read sequencing was performed after methylation differences were observed). These genes are BCLAF3, DIP2B and CSNK1E. The expansions have not been reported with DEE before (I do see that there has been 1 paper on DIP2B published over 15 years ago).

Further work will be needed to show that these are bone fide disease genes and will likely require methylation studies as a first step to identify patients who have an

underlying expansion – followed by expansion detection. The investigation into the DMRs (differentially methylated regions) also highlighted the ability for the analysis to identify copy number variants and complex rearrangements.

Of particular interest, was a screen of the 516 unsolved with the known 70 validated signatures (via EpiSign). The known signatures identified genetic diagnoses in 1% in the undiagnosed cohort. When combined with the assessment of rare, outlier DMRs the solve rate was about 2%.

While the overall solve rate was low, this was a sample that had previously undergone extensive testing. If the methylation assay had been performed earlier in the diagnostic journey, the results would have likely shown a much higher “solve rate”.

The researchers also perform further studies in 29 individuals with CHD2 variation and further refine an already-known methylation signature in DEE94.

The researchers appropriately comment on a major limitation, that the most common causes of DEE are under-represented with known, validated episignatures. They state that the only validated signature in DEE is CHD2 (and hence they studied this further).

This is an important limitation though, in time, other signatures will be validated for the DEE's.

Discussion is well written and balanced and discusses the benefits and challenges for this emerging technology in rare disease.

Supplemental is well written and I have no concerns.

We thank the reviewer for this comment and appreciate the recognition of this effort. We have since made some minor changes to the phenotype descriptions for continuity across individuals. These changes do not actually change the descriptions themselves and are, therefore, not highlighted.

Major comments

I have none. This is an important manuscript that will important for all individuals studying the underlying causes of epilepsy and encephalopathies. Methylation will be an important part of the diagnostic process for the unsolved.

We thank the reviewer for the positive feedback on our manuscript and for reaffirming the importance of this study in the current and future landscape of diagnostic testing.

Minor comments only

Introduction: Can the authors provide a reference for the sentence, “This, in turn, improves outcomes but is not possible when the etiology is unknown (“unsolved”).

We have added citations to support this statement that the outcome for DEEs is improved when the genetic etiology is known. These citations include [Line 131]:

- Kohler et al., *EJHG* 2017 (PMID: 28295040)
- Jeffrey et al., *Epilepsia Open* 2021 (PMID: 33681658)
- Swartwood et al., *Epilepsia Open* 2023 (PMID: 3807147)

Introduction: Can the authors add reference for “Episignatures have been found for neurodevelopmental disorders where epilepsy is part of the phenotype....”

We have added citations for examples of episignatures where epilepsy or seizures are part of the phenotype. The cited genes, episignatures, and disorders include [Line 163]:

- *CHD2*: Developmental and epileptic encephalopathy 94 (MIM:615369)
 - Aref-Eshghi et al., *AJHG* 2020 (PMID: 32109418),
- *KDM5C*: Intellectual developmental disorder, X-linked syndromic, Claes-Jensen type (MIM:300534)
 - Aref-Eshghi et al., *AJHG* 2020 (PMID: 32109418)
- *KDM2B*: MIM609078
 - Jaarsveld et al., *GIM* 2023 (PMID: 36322151)
- *SETD1B*: Intellectual developmental disorder with seizures and language delay (MIM:619000)
 - Aref-Eshghi et al., *AJHG* 2020 (PMID: 32109418)
- *KMT2A*: Wiedemann-Steiner syndrome (MIM: 605130)
 - Foroutan et al., *Int J Mol Sci* 2022 (PMID: 35163737)
- *SMARCA2*: Blepharophimosis-intellectual disability syndrome (MIM:619293)
 - Cappuccio et al., *GIM* 2020 (PMID: 32694869)
- *TET3*: Beck-Fahrner syndrome (MIM:618798)
 - Levy et al., *Genomic Medicine* 2023 (PMID: 34750377)
- *UBE2A*: Intellectual developmental disorder, X-linked syndromic, Nascimento type (MIM:300860)
 - Aref-Eshghi et al., *AJHG* 2020 (PMID: 32109418)

Material and methods: Can you provide numbers with the percent, 80% had a gene panel, 40% microarray analysis, 76% ES, and 38% GS. Collectively, 98% had at least 193 one sequence-based investigation (gene panel, ES, or GS)

We have updated this section to include the 66 additional patients we have added to this revision, and we now include the numbers corresponding to the percentages listed.

[Lines 196-200]: “After quality control and normalization (described below), there were 582 remaining individuals with unsolved DEEs who had undergone extensive molecular testing: 79% (458 individuals) had a gene panel, 51% (298 individuals) microarray or karyotype analysis, 75% (435 individuals) ES, and 40% (232 individuals) GS. Collectively, 97% (562 individuals) had at least one sequence-based investigation (gene panel, ES, or GS).”

Results: Can they include the number of repeats detected by LRS for the variants in CSNK1E, DIP2B and BCLAF3

Long-read sequencing provides an estimated range for the number of repeats, which can be viewed by examining the insertions evident in the data visualized in IGV. The number of repeats determined by LRS for *CSNK1E* was included in Supplementary Figure 8 but is now also noted in the main text.

[Lines 538-546]: “After validation of hypermethylation with targeted EM-seq for both probands (Supplementary Figure 5), long-read sequencing of the proband (genome, ~1,500-3,000bp) and mother (targeted, ~1,500bp) from Family 1 and the proband from Family 2 (genome, ~1,100-3,200bp) confirmed the presence of an expanded CGG motif in both (Figure 2B), as previously reported in individuals with hypermethylation of *CSNK1E* at fragile site *FRA22A* and reduced expression in lymphoblastoid cells (17). Through GeneMatcher (75), we identified Family 3 consisting of a proband with the same *CSNK1E* hypermethylated DMR and CGG repeat expansion (genome, ~1,300bp-2,100bp) inherited from his mother (genome, ~270bp-3,500bp), who is mildly affected by learning, speech, and sleep difficulties (Supplementary Phenotype data).”

The number of repeats determined by LRS for *DIP2B* is now included in the text:

[Lines 558-561]: “A male individual with unsolved DEE displayed maternally inherited hypermethylation of the *DIP2B* (MIM:611379) promoter region and exon 1 (Supplementary Figure 10), due to an underlying CGG-repeat expansion (~1,300-2,300bp), previously characterized as fragile site *FRA12A* (76).”

The number of repeats determined by LRS for *BCLAF3* was previously included in the text:

[Lines 566-570]: “We validated hypermethylation using targeted EM-seq (Supplementary Figure 5), and ONT long-read sequencing of the proband and his mother revealed a novel CGG repeat expansion in the proband (~2,500-3,000bp, Supplementary Figure 11) inherited from his mother, who had a smaller expansion (~1,700-1,900bp). LRS and standard X-inactivation studies (77).”

Line 641: “An additional 40 individuals with DEEs (80% unsolved) and....”. I suspect the 80% is a typo and should be 8% (40/516).

In this statement, the 80% refers to 80% of the 40 individuals with DEE (32/40) which are unsolved. The other 8 individuals with DEE are solved and used in the study as positive/negative controls. We have updated this sentence to clarify the numbers:

[Lines 691-693]: *“An additional 40 individuals with DEEs (n=32 unsolved, n=8 solved) and nine controls had inconclusive results for epesignatures...”*

We thank reviewer #2 for their encouraging comments and feedback. We have updated the manuscript to reflect the edits accordingly.

REVIEWERS' COMMENTS

Reviewer #2 (Remarks to the Author):

Thank you again for the opportunity to review this manuscript. I had only a few comments in the original draft and the authors have addressed them appropriately. Moreover, the authors have added additional 66 cases with similar solve rates (SMS and STX1B-TSS). I have no further comments.

Reviewer #3 (Remarks to the Author):

This study is novel as it shows that genome-wide DNA methylation analyses can be used to help understanding neurodevelopmental disorders with epilepsy. The authors conducted several investigations from a large group of patients and healthy controls. They were able to explain 2% of the unsolved cases. Although this seems a small improvements, the study is scientifically useful to the field and it can be an example for conducting similar studies in other patients. The authors have already revised the manuscript following reviewers' suggestions and added more analyses which have improved the paper. I would advise to address these comments before publication:

- 1) Lines 137-139: although some text has been added, it is still not clear what they mean by stating that epivariants are part of human genetic variation. More clarification on accepted definitions of epivariants and their relationship with genetic variants and with epigenetic modification is needed, included at least one reference.
- 2) Line 142-143 need a reference to the study or the studies showing the role of methylation in fragile X syndrome.
- 3) The study design is quite complex and although a flowchart is shown in supplementary figure, this needs to be in the main paper.
- 4) The various investigations that have been conducted should be summarised in a more straightforward way, not just the main analysis in Supplementary Figure 1A, but also all the follow up and validations that have been conducted need to be summarised in the context of the broader paper, for instance with a paragraph introducing all the analyses at the start of the Methods and a paragraph summarising all the results at the start of the Discussion. This would improve readability for non-specialists.
- 5) The batch structure of the arrays is not clear. In Suppl. Table 1A and at lines 218-220, it states that a total of 1224 individuals were had methylation measured across different batches. However, at lines 184 it mentioned 593 unsolved DEEs individuals and 475 healthy controls. Some of the controls were from a public database and other from an institutional database. Does it mean that these had been run previously in different labs? Differences between unsolved DEEs and controls could then be due to a strong batch effect deriving from this procedure. This is particularly problematic in a study based on looking at rare differences, in some cases on 1-2 individuals where there is not enough replication across individuals. Can the authors be more specific about this batch issue and confirm the validity of the results via some replicative methods? If they have already done this, can they make it clearer across the paper?
- 6) Were the healthy controls matched to the DEEs by sex and age?
- 7) Some of the figures are quite complex and with very small text. Could the authors try to improve their interpretability? For instance, Figure 4 is not legible.

REVIEWERS' COMMENTS

Our responses are in blue and changes made to the text of the manuscript relevant to our responses are in red.

Reviewer #2 (Remarks to the Author):

Thank you again for the opportunity to review this manuscript. I had only a few comments in the original draft and the authors have addressed them appropriately. Moreover, the authors have added additional 66 cases with similar solve rates (SMS and STX1B-TSS).

I have no further comments.

We thank reviewer #2 for their time and input towards improving this manuscript.

Reviewer #3 (Remarks to the Author):

This study is novel as it shows that genome-wide DNA methylation analyses can be used to help understanding neurodevelopmental disorders with epilepsy. The authors conducted several investigations from a large group of patients and healthy controls. They were able to explain 2% of the unsolved cases. Although this seems a small improvements, the study is scientifically useful to the field and it can be an example for conducting similar studies in other patients. The authors have already revised the manuscript following reviewers' suggestions and added more analyses which have improved the paper.

We thank reviewer #3 for their helpful feedback on our paper. We have addressed each of the following areas below.

I would advise to address these comments before publication:

1) Lines 137-139: although some text has been added, it is still not clear what they mean by stating that epivariants are part of human genetic variation. More clarification on accepted definitions of epivariants and their relationship with genetic variants and with epigenetic modification is needed, included at least one reference.

The reviewer makes a good point that additional clarification and citations are needed to support the definition of epivariants. We have added clarity about the definition of epivariants to:

Lines 133-135: "Rare epivariants, defined as rare alterations in DNA methylation with or without identified underlying DNA sequence alterations, contribute to human genetic variation [17], but have also been shown to disrupt normal methylation and transcription to cause disease [18,19]."

We have also added citations:

- To support the claim that rare epivariations are a part of normal human variation, we now cite Garg et al., 2020 AJHG (PMID:32937144). This study interrogated the prevalence and distribution of rare epivariations in the human population (n>23,000 presumably healthy controls). They described >4,000 unique epivariations.
- To support the claim that rare epivariations cause disease, we have added citations for hypermethylation of *MSH2* in Lynch syndrome (PMID:19098912) and *BRCA1* in breast cancer (PMID: 30075112), both of which are associated with underlying DNA variants.

2) Line 142-143 need a reference to the study or the studies showing the role of methylation in fragile X syndrome.

We now cite a recent review that summarizes the distinct molecular mechanism of Fragile X, with an emphasis on the role of DNA methylation.

Lines 142-144: “One example is the methylation of **CGG repeats** in the 5’ untranslated region (5’UTR) of *FMR1* (MIM:309550) that represses gene expression and causes Fragile X syndrome [20] (MIM:300624).”

3) The study design is quite complex and although a flowchart is shown in supplementary figure, this needs to be in the main paper.

We have moved Supplementary Figure 1 to the main manuscript as Figure 1. We have also added an additional graphical schematic (see below response to #4) to this figure to aid in the understanding of our study design. We hope that this has improved the clarity of the methods and cohort.

4) The various investigations that have been conducted should be summarised in a more straightforward way, not just the main analysis in Supplementary Figure 1A, but also all the follow up and validations that have been conducted need to be summarised in the context of the broader paper, for instance with a paragraph introducing all the

analyses at the start of the Methods and a paragraph summarising all the results at the start of the Discussion. This would improve readability for non-specialists.

We thank the reviewer for this comment about providing more summaries describing the methods in the manuscript. We have now added a graphical schematic (new Figure 1) to the main document, providing a concise description of the methods and findings.

5) The batch structure of the arrays is not clear. In Suppl. Table 1A and at lines 218-220, it states that a total of 1224 individuals were had methylation measured across different batches. However, at lines 184 it mentioned 593 unsolved DEEs individuals and 475 healthy controls. Some of the controls were from a public database and other from an institutional database. Does it mean that these had been run previously in different labs? Differences between unsolved DEEs and controls could then be due to a strong batch effect deriving from this procedure. This is particularly problematic in a study based on looking at rare differences, in some cases on 1-2 individuals where there is not enough replication across individuals. Can the authors be more specific about this batch issue and confirm the validity of the results via some replicative methods? If they have already done this, can they make it clearer across the paper?

We have now clarified the batch structure by adding additional text for clarification in the methods (see below), an additional “Sample_Plate/Batch” column in the supplementary tables (Supplementary Table 1B and 1C), and a supplementary figure (new Supplementary Figure 1) displaying the PCA plot batch effect before and after correction using ComBat. As a validation, DMR outlier analysis was performed (1) with batch correction of SJNORM as displayed and (2) with batch correction accounting for every batch as shown in the figure legend. Method (1) was chosen as the best approach since method (2) resulted in the overcorrection and the loss of a significant disease-causing DMR with a robustly underlying DNA defect.

Lines 750-755: “Since samples were run in multiple batches and at different institutions, we visually examined the PCA plot for batch effects. The only batch effect observed was on PC1 between the SJLIFE unaffected control cohort and the rest of the samples analyzed (including both cases and controls). We used the SVA [65] for batch correction using the ComBat method and confirmed the elimination of the batch effect (Supplementary Figure 1, Supplementary Table 1B) [66].”

6) Were the healthy controls matched to the DEEs by sex and age?

Sex: Reported sex distributions between DEEs and controls are adequately matched/represented. These proportions are now mentioned in the main manuscript (Lines 719-723). Recall that predicted males and females are separated for the outlier DMR analysis of the sex chromosomes as described in lines 761-762. Additionally, chrX probes are excluded from episignature analyses.

Age:

- **Episignatures:** For signature testing and *CHD2* episignature refinement, the selection of controls from the EpiSign Knowledge Database are age and sex-matched, which is important for analyzing individual CpG probe sites.
- **Outlier DMRs:** Large cohorts of pediatric (age at sample collection < 18 years), presumably healthy, reference controls are not widely unavailable. Therefore, we used SJNORM healthy controls (adults), PPMI publicly available controls (adults), and unaffected parents (adults) or siblings (children) of probands. We employed these controls to filter down DMRs specific to individuals with DEEs.

Supplementary Figure 1. Correction of batch effects detected through PCA analysis.

Although we did not specifically age match cases and controls for the outlier DMR analysis:

- **#1** Since Robust, rare differentially methylated regions (DMRs) are frequently due to underlying DNA variants, they are expected to be stable across the individual's lifetime. Therefore, regardless of age, these controls should represent the purposes of filtering DMRs of interest in our DEE cohort.
- **#2** Rare outlier DMRs have been shown to accumulate with age (PMID:32937144). Since these events increase over time, we are more likely to bias ourselves toward more stringent filtering of outlier DMRs in cases (largely pediatric) vs. controls (largely adults). This would make our findings more robust (i.e. DMRs are present in cases and never in controls).
- Also, we confirm that age did not create a considerable batch effect in the PCA plot, and the DMRs we reported in this manuscript were all validated with an orthogonal approach.

7) Some of the figures are quite complex and with very small text. Could the authors try to improve their interpretability? For instance, Figure 4 is not legible.

We have improved the readability of Figure 4 and the other figures to ensure they are legible.